# Lactate receptor HCAR1 regulates neurogenesis and microglia activation after neonatal hypoxia-ischemia

Lauritz Kennedy[1,2†], Emilie R Glesaaen[1,2†], Vuk Palibrk[3], Marco Pannone[1,3], Wei Wang[3], Ali Al-Jabri[1,2], Rajikala Suganthan[1], Niklas Meyer[1,2], Marie Landa Austbø[1], Xiaolin Lin[1,3], Linda H Bergersen[4,5], Magnar Bjørås[1,3‡], Johanne E Rinholm[1,2*‡]

[1]Department of Microbiology, Oslo University Hospital and University of Oslo, Oslo, Norway; [2]Division of Physiology, Institute of Basic Medical Sciences, University of Oslo, Oslo, Norway; [3]Department of Clinical and Molecular Medicine, Norwegian University of Science and Technology, Trondheim, Norway; [4]The Brain and Muscle Energy Group, Institute of Oral Biology, Faculty of Dentistry, University of Oslo, Oslo, Norway; [5]Center for Healthy Aging, Department of Neuroscience and Pharmacology, Faculty of Health Sciences, University of Copenhagen, Copenhagen, Denmark

**Abstract** Neonatal cerebral hypoxia-ischemia (HI) is the leading cause of death and disability in newborns with the only current treatment being hypothermia. An increased understanding of the pathways that facilitate tissue repair after HI may aid the development of better treatments. Here, we study the role of lactate receptor HCAR1 in tissue repair after neonatal HI in mice. We show that HCAR1 knockout mice have reduced tissue regeneration compared with wildtype mice. Furthermore, proliferation of neural progenitor cells and glial cells, as well as microglial activation was impaired. Transcriptome analysis showed a strong transcriptional response to HI in the subventricular zone of wildtype mice involving about 7300 genes. In contrast, the HCAR1 knockout mice showed a modest response, involving about 750 genes. Notably, fundamental processes in tissue repair such as cell cycle and innate immunity were dysregulated in HCAR1 knockout. Our data suggest that HCAR1 is a key transcriptional regulator of pathways that promote tissue regeneration after HI.

*For correspondence:
j.e.rinholm@medisin.uio.no

†These authors contributed equally to this work
‡These authors also contributed equally to this work

## Editor's evaluation

This manuscript highlights the contribution of lactate receptor HCAR1 to mechanisms of neuronal repair after hypoxic injury. This paper will be of interest to scientists studying mechanisms of hypoxia-ischemia-induced brain injury and tissue repair and has high translational relevance. It implicates a novel mechanism, lactate-HCAR1 signaling, as underlying cellular proliferation and neurogenesis needed for tissue regeneration after injury. A series of compelling experiments presented in this article support the necessary role of HCAR1 in these effects.

## Introduction

Cerebral hypoxia-ischemia (HI) affects around 1.5 per 1000 live born births in the developed countries (*Kurinczuk et al., 2010*). It is characterised by an insufficient supply of blood and oxygen to the brain, leading to cell death and brain tissue damage. Hypothermia is the mainstay of today's treatment (*Shankaran et al., 2012*), but a high percentage of survivors still experience long-term neurological effects, including cerebral palsy, epilepsy and cognitive disabilities (*Douglas-Escobar and Weiss,*

**eLife digest** Hypoxic-ischaemic brain injury is the most common cause of disability in newborn babies. This happens when the blood supply to the brain is temporarily blocked during birth and cells do not receive the oxygen and nutrients they need to survive. Cooling the babies down after the hypoxic-ischemic attack (via a technique called hypothermic treatment) can to some extent reduce the damage caused by the injury. However, doctors still need new drugs that can protect the brain and improve its recovery after the injury has occurred.

Research in mice suggests that a chemical called lactate might help the brain to recover. Lactate is produced by muscles during hard exercise to provide energy to cells when oxygen levels are low. Recent studies have shown that it can also act as a signalling molecule that binds to a receptor called HCAR1 (short for hydroxycarboxylic acid receptor) on the surface of cells. However, it is poorly understood what role HCAR1 plays in the brain and whether it helps the brain recover from a hypoxic-ischaemic injury.

To investigate, Kennedy et al. compared newborn mice with and without the gene that codes for HCAR1 that had undergone a hypoxic-ischaemic brain injury. While HCAR1 did not protect the mice from the disease, it did help their brains to heal. Mice with the gene for HCAR1 partly recovered some of their damaged brain tissue six weeks after the injury. Their cells switched on thousands of genes involved in the immune system and cell cycle, resulting in new brain cells being formed that could repopulate the injured areas. In contrast, the brain tissue of mice lacking HCAR1 barely produced any new cells.

These findings suggest that HCAR1 may help with brain recovery after hypoxia-ischemia in newborn mice. This could lead to the development of drugs that might reduce or repair brain damage in newborn babies. However, further studies are needed to investigate whether HCAR1 has the same effect in humans.

*2015*). Following a hypoxic-ischemic event, the neonatal brain has the ability to partly regenerate (*Donega et al., 2013*). This process of brain tissue regeneration requires a coordinated increase in microglia-induced inflammation, cell proliferation and angiogenesis. After the acute phase of cerebral HI, debris from dead cells activates microglia, the resident immune cells of the brain. These cells have the ability to remove cell debris by phagocytosis and release factors that stimulate tissue repair (*Jin and Yamashita, 2016*). At the same time, the hypoxic-ischemic event leads to the release of various growth factors, which stimulate proliferation of neural progenitor cells (*Donega et al., 2013*) as well as angiogenesis (*Donega et al., 2013*; *Shweiki et al., 1992*). In mammalian postnatal brain, the majority of progenitor cells are located in proliferating areas of the brain, of which the most well described are the subventricular zone, located adjacent to the lateral ventricles, and the dentate gyrus of the hippocampus. Proliferating cells migrate from these areas to repopulate the damaged tissue. Targeting these areas is therefore a potential strategy to stimulate repair processes after an ischemic insult.

Recent studies in mice have shown improved recovery after neonatal HI by administration of lactate (*Roumes et al., 2021*; *Tassinari et al., 2020*), but the underlying mechanisms for this beneficial effect are unclear. It is unknown whether lactate improves recovery by giving metabolic support to the cells or by signalling via the Hydroxycarboxylic acid receptor 1 (HCAR1), or both.

HCAR1 is a $G_i$-protein-coupled receptor. It was first described in adipose tissue where it inhibits lipolysis through lowering of cyclic adenine monophosphate (cAMP; *Ahmed et al., 2010*). In the brain, HCAR1 activation can modulate neuronal firing rates in vitro (*de Castro Abrantes et al., 2019*) and stimulate brain angiogenesis in vivo (*Morland et al., 2017*). Until now, a role of HCAR1 in tissue protection or repair after ischemia has not been demonstrated, although studies from different cell lines suggest that it may regulate cell proliferation and differentiation (*Stäubert et al., 2015*; *Wu et al., 2018b*).

Here, we investigate the role of HCAR1 in neonatal HI in mice. We show that HCAR1 knockout (KO) mice have a reduced ability to regenerate brain tissue after HI. By examination of neurosphere cultures in vitro and immunohistochemical staining of brain tissue after HI, we find that HCAR1 KO mice display impaired proliferation of neural progenitor cells. Furthermore, HI-reactive proliferation of microglia, astrocytes and oligodendrocyte progenitors is perturbed in the HCAR1 KO. In addition, we

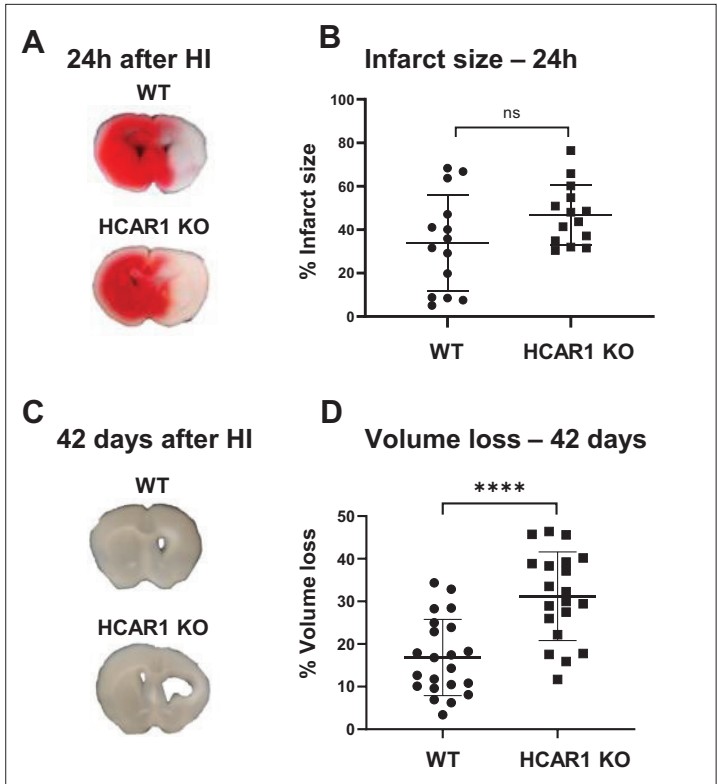

**Figure 1.** HCAR1 does not protect the brain from acute tissue damage following cerebral HI, but promotes brain tissue regeneration. (**A**) Representative images of TTC-stained brain sections from WT and HCAR1 KO mice 24 hr after HI. TTC turns red upon reacting with mitochondrial respiratory enzymes in viable tissue, whereas infarcted tissue remains white. (**B**) Brain infarct size (TTC-negative tissue as percentage of total quantified tissue volume) 24 hr after HI. p=0.19, n=14 mice/genotype. (**C**) Representative images of coronal brain sections from WT and HCAR1 KO mice 42 days after HI. (**D**) Brain tissue loss (% of total quantified tissue volume) 42 days after HI. p=2.2*10$^{-5}$, WT n=22, KO n=20. Error bars indicate SD. Statistical significance was calculated using a two-tailed t-test.

find that the microglia are less activated post HI. Transcriptome analysis revealed that subventricular zones from HCAR1 KO mice have an almost complete lack of transcriptional response to HI. This was specific to the subventricular zone as hippocampal samples from HCAR1 KO mice responded similar to that of wildtype (WT) mice. Thus, HCAR1 is a crucial transcriptional regulator of tissue response to ischemia in the subventricular zone. HCAR1 could therefore be targeted to promote tissue repair after HI.

## Results
### HCAR1 is required for brain tissue regeneration after HI

To investigate the role of HCAR1 in stress-induced neuronal injury and subsequent neurogenesis, we induced HI in 9 days old HCAR1 KO and WT mice. Developmentally, this age in mice is thought to represent the human infant at term (*Hagberg et al., 2002*). We used a model for cerebral HI that includes permanent occlusion of the left common carotid artery followed by systemic hypoxia (*Sejersted et al., 2011*). This leads to a detectable histological injury in the cortex, hippocampus, striatum, and thalamus of the left hemisphere, whereas the contralateral hemisphere is indistinguishable from a sham-treated brain, constituting a morphologically accurate internal control (*Sejersted et al., 2011*). After HI, we examined acute brain tissue damage and long-term tissue loss. We assessed acute brain tissue damage by 2,3,5-Triphenyltetrazolium chloride (TTC) staining. HCAR1 KO and WT mice showed comprehensive damage in the affected ipsilateral hemisphere 24 hr after HI, with an average TTC-negative (i.e. damaged) volume in the ipsilateral relative to the contralateral side of 34% ± 22% in WT

and 47% ± 14% in HCAR1 KO. There was no significant difference between HCAR1 KO and WT mice in total acute tissue damage (*Figure 1A–B*, p=0.19).

After the acute phase of HI, there is a phase of neurogenesis and tissue repair. To assess a potential role for HCAR1 in tissue repair, we measured tissue loss 42 days after HI. At this time point, the repair process is completed and the long-term damage can be observed as a loss of brain tissue (*Sejersted et al., 2011*). WT mice showed partial restoration of damaged brain structures with a tissue loss of 17% ± 9% (*Figure 1C–D*). In comparison, HCAR1 KO mice showed significantly more tissue deficit with a permanent tissue loss of 31% ± 10%, that is 82% higher than in WT mice (*Figure 1D–E*, p=2.2*10$^{-5}$). This suggests that HCAR1 is important for induced neurogenesis and tissue repair after HI.

## Impaired proliferation of neural progenitor cells in HCAR1 KO mice

Tissue repair after an ischemic injury is aided by an increase in proliferation and differentiation of neural progenitor cells (*Lindvall and Kokaia, 2015*; *Mattiesen et al., 2009*; *Plane et al., 2004*). To test the effect of HCAR1 on neural progenitor proliferation and cell fate, we performed a neurosphere assay on spheres derived from forebrains of HCAR1 KO and WT mice. Neurospheres offer a simplified and isolated in vitro system where proliferation, self-renewal, and differentiation can be tested in a controlled environment. We found that neurospheres from HCAR1 KO mice developed fewer colonies compared with neurospheres from WT mice (*Figure 2A–B and G*, no of colonies per well WT 178±23; KO 132±10, p=0.034). This indicates that HCAR1 KO neural progenitors have a lower self-renewal and less proliferation. The average size of the spheres was not significantly different between the two genotypes (*Figure 2A–B and H*, sphere area WT 289±13 µm$^2$; KO 250±21 µm$^2$, p=0.053). To examine whether HCAR1 can affect cell fate, the neurospheres were dissociated, and the cells were cultured in differentiation medium for 5 days and immunolabelled with the neuronal marker Tuj1 and the astrocyte and neural progenitor marker GFAP. We detected Tuj1 + cells as well as GFAP + cells after induced differentiation. However, HCAR1 KO cells had a lower percentage of Tuj1 +neurons compared with WT cells (*Figure 2C–D,I*, WT Tuj1 28% ± 2%; KO Tuj1 19% ± 2%, p<0.001), whereas the percentage of GFAP + cells was not significantly different between the two genotypes (*Figure 2E–F,I*, WT GFAP 18% ± 3%; KO GFAP 22% ± 3%, p=0.08). This suggests that HCAR1 may direct progenitor cells towards a neuronal fate, although more studies are needed to very an effect on cell differentiation.

As the neurosphere data suggested impaired proliferation of neural progenitors, we then examined in vivo cell proliferation after HI in HCAR1 KO and WT mice. Mice were injected with the proliferation marker Bromodeoxyuridine (BrdU) on days 4–7 after HI and were sacrificed on day 7 (method adapted from *Hayashi et al., 2005*; *Palibrk et al., 2016*; *Plane et al., 2004*). The density of BrdU + cells was assessed by immunohistochemistry on brain sections. Our analysis focused on the striatal subventricular niche as this is an area containing a large portion of neural progenitor cells that undergo cell proliferation post hypoxic ischemia (*Hayashi et al., 2005*; *Plane et al., 2004*). As there has been observed regional differences in the HI-response of the SVZ (*Sejersted et al., 2011*), we analysed two separate areas of the subventricular niche, namely the ventricular adjoining ventricular-subventricular zone (V-SVZ) and the more lateral subventricular-intermediate zone (SVZ-IZ). Since the Dentate gyrus of the Hippocampus also contains a high number of neuronal progenitor cells, we wanted to examine cell proliferation in this area too. Unfortunately, on day 7 after HI, the hippocampus was mostly completely degenerated on the ipsilateral side in most of the mice (not shown). In the SVZ-IZ the overall cell density was increased by 47% in the HI affected ipsilateral hemisphere compared with the contralateral hemisphere in WT mice (*Figure 2K–L and O*, DAPI + cells/0.01 mm$^2$: contra 30.9±4.0, ipsi 45.6±4.3, p<0.001). In KO mice, however, there was no change in cell density (*Figure 2M–O*, KO DAPI + cells/0.01 mm$^2$: contra 35.0±7.3, ipsi 34.3±5.4, p=0.95). WT mice also had a significant increase in the density of proliferated BrdU + cells on the ipsilateral side, but this was not the case for HCAR1 KO mice (*Figure 2—figure supplement 1A*). Since overall cell density (determined by DAPI) varied somewhat between mice within the same experimental group (*Figure 2O*), we also looked at the BrdU/DAPI ratio to determine whether the ratio of BrdU cells had increased after HI. WT mice had doubled the ratio of proliferated BrdU + cells on the ipsilateral side (*Figure 2K–L and P*, %BrdU+/DAPI + cells: contra 6.9±2.0, ipsi 17.1±3.2, p<0.001), while there was no significant change between ipsi and contralateral sides in HCAR1 KO (%BrdU+/DAPI + cells: contra 9.5±3.7, ipsi 11.8±4.3, p=0.15). Together, these data suggest that HCAR1 KO mice are unable to fully instigate cell proliferation after HI.

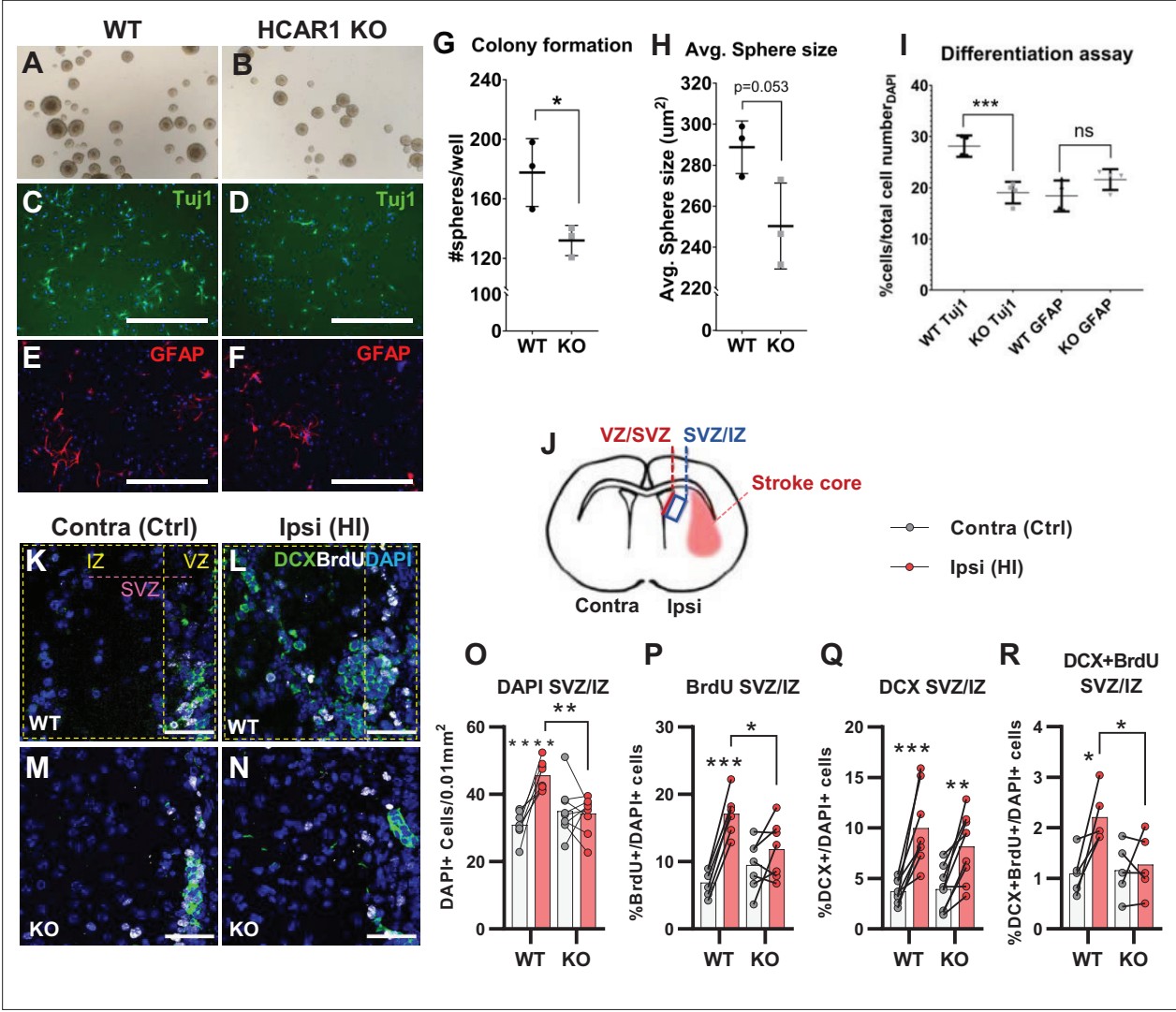

**Figure 2.** HCAR1 regulates neural progenitor cell proliferation. A-I Neurosphere formation from HCAR1 KO and WT cells. (**A–B**) Images of neurospheres from WT (**A**) and HCAR1 KO (**B**) mice. (**C–F**) Fluorescence images from WT (**C, E**) and HCAR1 KO (**D, F**) dissociated neurospheres after induced differentiation. Scale bar: 400 µm. (**C–D**) are stained with the neuronal marker Tuj1 and (**E–F**) with the astrocyte and neural progenitor marker GFAP. (**G**) Number of colonies formed per well, p=0.034, df 4, n=3 clones/genotype. (**H**) Size of neurospheres (um²), p=0.053, df 4, n=3. (**I**) Percentage of cells positive for Tuj1 or GFAP after induced differentiation. Tuj1 KO vs WT p=0.0009; GFAP KO vs WT p=0.08, df 12, n=4 clones/genotype. Data in (**G–I**) are shown as mean ± SD and p-values were calculated with two-tailed t-test. (**J–R**) Quantification of proliferated cells and neuronal progenitors after HI. (**J**) Illustration of a coronal mouse brain section indicating the core of the infarct in the ipsilateral hemisphere (Ipsi), the contralateral hemisphere (contra, used as control) as well as the ventricular-subventricular sone (V-SVZ) and subventricular-intermediate zone (SVZ-IZ). (**K–N**) Confocal images from coronal mouse brain sections labelled for DAPI (blue), doublecortin (DCX, marker of neuronal progenitor cells, green) and BrdU (injected proliferation marker, white). The images show the V-SVZ (VZ) and SVZ-IZ (IZ) zones in the contralateral (**K, M**) and ipsilateral (**L, N**) hemispheres in WT (**K–L**) and KO (**M–N**) mice. The purple line indicates the SVZ. Scale bars: 50 µm. (**O–R**) Density (**O**) or ratio (**P–R**) of cells in the SVZ-IZ zones of the ipsi- (pink bars) and contralateral (white bars) hemispheres of WT and KO mice. (**O**) DAPI +nuclei (i.e. all cells), WT contra vs ipsi p<0.001, df 15, n=8. KO contra vs ipsi p=0.95, df 15, n=9. (**P**) Ratio of BrdU + cells (i.e. proliferated cells / DAPI), WT contra vs ipsi p<0.001, df 11, n=6. KO contra vs ipsi p=0.15, df 7, n=11. (**Q**) DCX + cells (neuronall progenitor cells/ DAPI), WT contra vs ipsi p<0.001, df 15, n=8. KO contra vs ipsi, p=0.002, df 15, n=9. (**R**) Ratio of proliferated neuronal progenitor cells (DCX +and BrdU+ / DAPI). WT contra vs ipsi p=0.01, df 11, n=6. KO contra vs ipsi p=0.92, df 11, n=7. Each point represents one sample/mouse. In (**O–R**) ipsi- and contralateral samples from the same mouse are indicated with a line. p-Values were calculated with Šídak method for multiple comparisons of selected groups after significant one-way ANOVA test.

The online version of this article includes the following figure supplement(s) for figure 2:

**Figure supplement 1.** Density data from subventricular-intermediate zone (SVZ/IZ) and quantification proliferated cells and neuroblasts in the ventricular-subventricular zone (VZ/SVZ) after HI.

We then looked at HI-induced proliferation of doublecortin positive (DCX+) neuronal progenitor cells. After HI the ratio of DCX+ / DAPI + cells more than doubled in both genotypes in the ipsilateral SVZ-IZ indicating intact reactive neurogenesis in the HCAR1 KO mice (*Figure 2K–N and Q*. %DCX+/ DAPI + cells: WT contra 3.7±1.2, ipsi 10.0±3.8, p<0.001. KO contra 3.9±2.1, ipsi 8.2±3.8, p=0.002). However, the ratio of BrdU +neuronal progenitors was only increased in WT mice (*Figure 2K–N and R*. %DCX +BrdU + /DAPI + cells: WT contra 1.1±0.4, ipsi 2.2±0.5, p=0.01. KO contra 1.2±0.5, ipsi 1.3±0.6, p=0.92). Similarly, the density of proliferating neuronal progenitors was only increased in WT, but not in HCAR1 KO mice after HI (*Figure 2—figure supplement 1B-C*). In the V-SVZ, there were no significant effect of HI or differences between the genotypes (*Figure 2—figure supplement 1D-G*). In conclusion, these data indicate that HCAR1 KO mice fail to increase proliferation of neural progenitor cells after HI, suggesting that HCAR1 is required to stimulate neuronal cell proliferation to induce tissue repair after ischemic damage.

## Impaired microglial proliferation and activation in HCAR1 KO mice after HI

After cerebral HI, microglia are recruited to the injured site at which they remove debris from dead cells to facilitate the repair process (*Neumann et al., 2009*). This requires increased proliferation, activation, and migration of the microglia (*Amat et al., 1996*; *Neumann et al., 2009*). We used immunohistochemistry to assess the proliferation and activation of microglia in the area surrounding the infarct (the peri-infarct zone). In contrary to the subventricular niche where only WT mice had an increase in overall cell density after HI, both genotypes had an increase in cell density in the peri-infarct zone (*Figure 3A–E*. DAPI + cells/0.01 mm$^2$: WT contra 25.2±3.8, ipsi 29.4±5.7, p=0.03. KO contra 23.5±4.0, ipsi 27.5±4.8, p=02). As expected, WT mice had a strong increase in the ratio of proliferating microglia in the ipsilateral hemisphere when compared with the contralateral side. However, no increase was detected in HCAR1 KO mice (*Figure 3A–D and G*. %IBA1 +BrdU + /DAPI + cells: WT contra 1.8±0.9, ipsi 3.9±2.4, p=0.02. KO contra 1.9±0.9, ipsi 1.8±1.1, p=0.99). Further, the ratio of microglia was increased in the ipsilateral side in WT, but not significantly in HCAR1 KO mice (*Figure 3A–D and F*. %IBA1+/DAPI + cells: WT contra 6.4±1.9, ipsi 11.5±5.8, p=0.005. KO contra 5.4±1.6, ipsi 7.9±2.1, p=0.12). These findings were also reflected in the density measurements of microglia in the same area (*Figure 3—figure supplement 1*).

We then assessed the activation of microglia. Activated microglia have larger cell soma and are less ramified (i.e. they have shorter and fewer branches) (*Morrison and Filosa, 2013*). In WT mice, microglia in the ipsilateral hemisphere had larger somata and were less ramified than in the contralateral side, indicating an activated phenotype (*Figure 3A–B,H-I*). Cross-sectional area, μm$^2$: WT contra 152.0±22.8, ipsi 176.5±17.5, p=0.02. Ramification (avg. max branch length, μm): WT contra 17.8.1±1.7, ipsi 16.1±0.9, p=0.01. No significant changes in the cell soma size and ramification of microglia were observed in HCAR1 KO mice (*Figure 3C–D,H-I*), cross-sectional area, μm$^2$: KO contra 148.1±14.0, ipsi 158.2±23.3, p=0.37. Ramification (Avg. max branch length, μm): KO contra 17.0±2.1, ipsi 16.3±1.5, p=0.36, indicating that microglia activation is not induced in the peri-infarct area by HI. In sum, these data suggest that HCAR1 KO mice were unable to initiate microglia proliferation and activation in response to HI, indicating a role for HCAR1 in these processes.

## Reactive astrogliosis and oligodendrocyte proliferation in the peri-infarct zone

The data above propose that HCAR1 is important for proliferation of microglia as well as neurons after HI. We therefore wanted to determine whether HCAR1 is also involved in the proliferation of other brain cells such as astrocytes and oligodendrocytes. Immunolabelling with GFAP and the proliferation marker BrdU, showed that the ratio of GFAP + cells and proliferated GFAP + cells was increased in the ipsilateral hemisphere in WT mice, but it was not significantly increased in HCAR1 KO mice (*Figure 4A–F*, %GFAP+/ DAPI + cells: WT contra 12.3±3.6, ipsi 20.6±5.0, p<0.001. KO contra 12.8±3.2, ipsi 15.9±1.7, p=0.07. %GFAP +BrdU + /DAPI + cells: WT contra 0.28±0.21, WT ipsi 0.73±0.45, p=0.002. KO contra 0.5±0.19, KO ipsi 0.59±0.35, p=0.81). Similarly, the density of proliferating GFAP + cells was increased after HI in WT but not in HCAR1 KO mice (*Figure 4—figure supplement 1A-B*). Although GFAP can label non-reactive as well as reactive astrocytes, it should be noted that not all non-reactive astrocytes are GFAP positive and reactive astrocytes are characterized

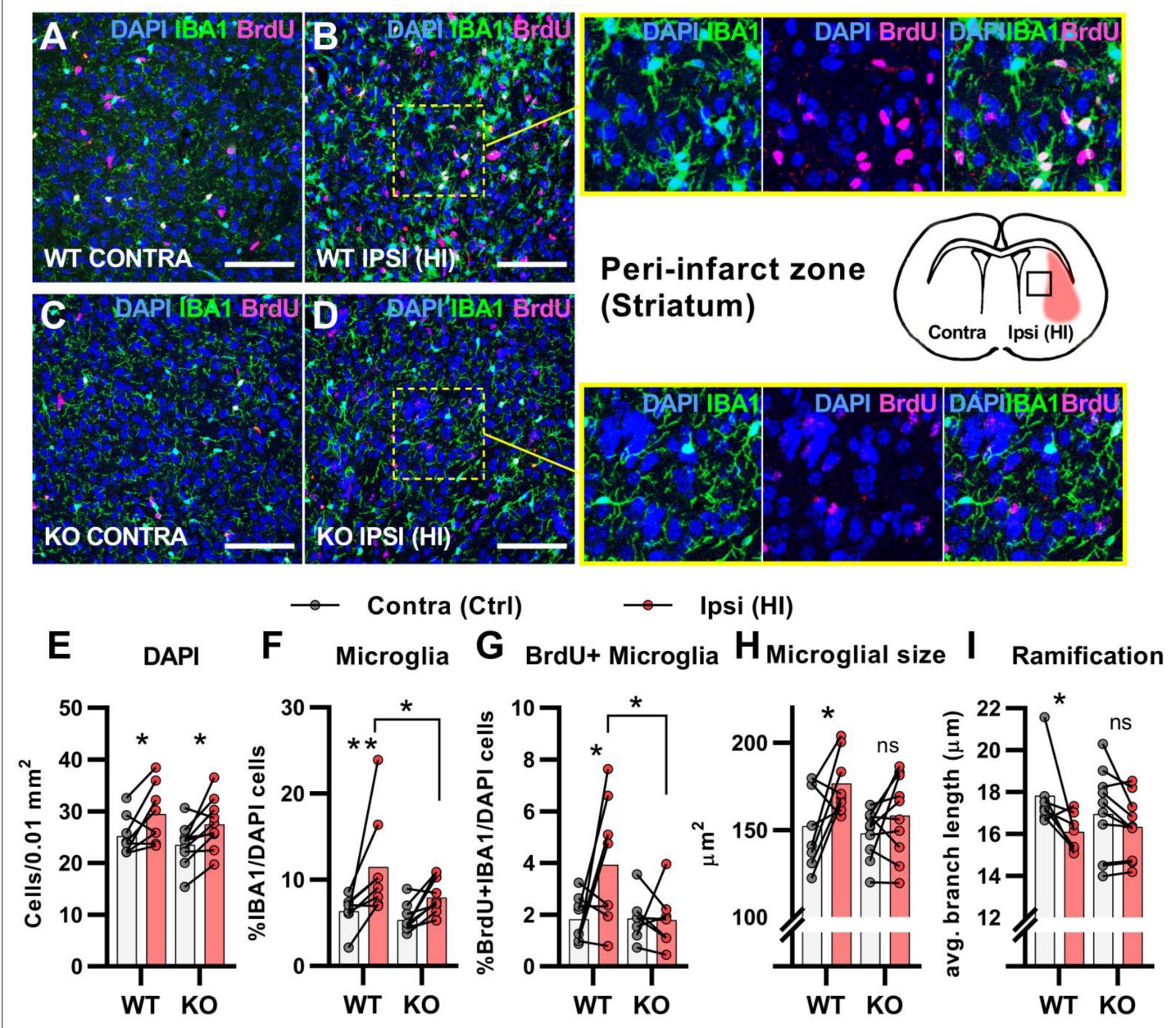

**Figure 3.** HCAR1 KO mice have deficient activation and proliferation of microglia after HI. (**A-D**) Confocal images from the peri-infarct zone (b, d, indicated as square in cartoon) and corresponding contralateral area (**A, C**) of coronal mouse brain sections from WT (**A–B**) and KO (**C–D**) labelled for BrdU (proliferating cells, magenta), Iba1 (microglia, green), and DAPI (blue nuclei). Scale bars: 100 μm (**E**) Density of all cells (DAPI+/0.01 mm$^2$) in the peri-infarct zone (pink bars) and contralateral striatum (white bars) of WT and KO mice. WT contra vs ipsi p=0.03, df 16, n=8. KO contra vs ipsi p=0.02, df 16, n=10. (**F**) Ratio of microglia (IBA1+/DAPI + cells) in the peri-infarct zone. WT contra vs ipsi p=0.005, df 16, n=8. KO contra vs ipsi p=0.12, df 16, n=10. (**G**) Ratio of proliferating microglia (i.e. cells that were both IBA1 +and BrdU+). WT contra vs ipsi p=0.02, df 26, n=8. KO contra vs ipsi p=0.99, df 26, n=7. (**H–I**) Assessment of microglia activation by morphology. When activated, microglia somata increase in size and get shorter and fewer branches. (**H**) Average size of microglia somata. WT contra vs ipsi p=0.02, df 16, n=8. KO contra vs ipsi p=0.37, df 16, n=10. (**I**) Average maximum branch length. WT contra vs ipsi p=0.01, df 15. n=7. KO contra vs ipsi p=0.36, df 15, n=10. Each point represents one sample/mouse. Ipsi- and contralateral samples from the same mouse are indicated with a line. p-Values were calculated with Šídák method for multiple comparisons of selected groups.

The online version of this article includes the following source code and figure supplement(s) for figure 3:

**Source code 1.** BrdU WEKA.

**Source code 2.** DAPI WEKA.

**Source code 3.** IBA1 WEKA.

*Figure 3 continued on next page*

*Figure 3 continued*

**Source code 4.** Script Microglia analysis.

**Figure supplement 1.** Density measurements of microglia in the peri-infarct zone.

by high GFAP levels. It is therefore likely that the increase in GFAP + cells primarily represent reactive astrogliosis, which is expected to occur in the peri-infarct area.

We then labelled oligodendrocytes with the marker Olig2, which stains oligodendrocytes at all developmental stages. The ratio of oligodendrocytes or proliferated oligodendrocytes in the peri-infarct zone did not change significantly in either HCAR1 KO or WT mice after HI (*Figure 4G–L*, %Olig2+/DAPI + cells: WT contra 30.3±7.7, ipsi 37.8±9.0, not tested with Sidak due to unsignificant one-way ANOVA test. KO contra 37.1±1.1, ipsi 31.6±11.2, not tested. %Olig2 +BrdU + /DAPI+: WT contra 3.3±1.8, ipsi 5.3±2.7, p=0.06. KO contra 3.2±1.1, ipsi 3.9±1.2, p=0.92). However, the oligodendrocyte density, and density of proliferated oligodendrocytes was increased in the ipsilateral

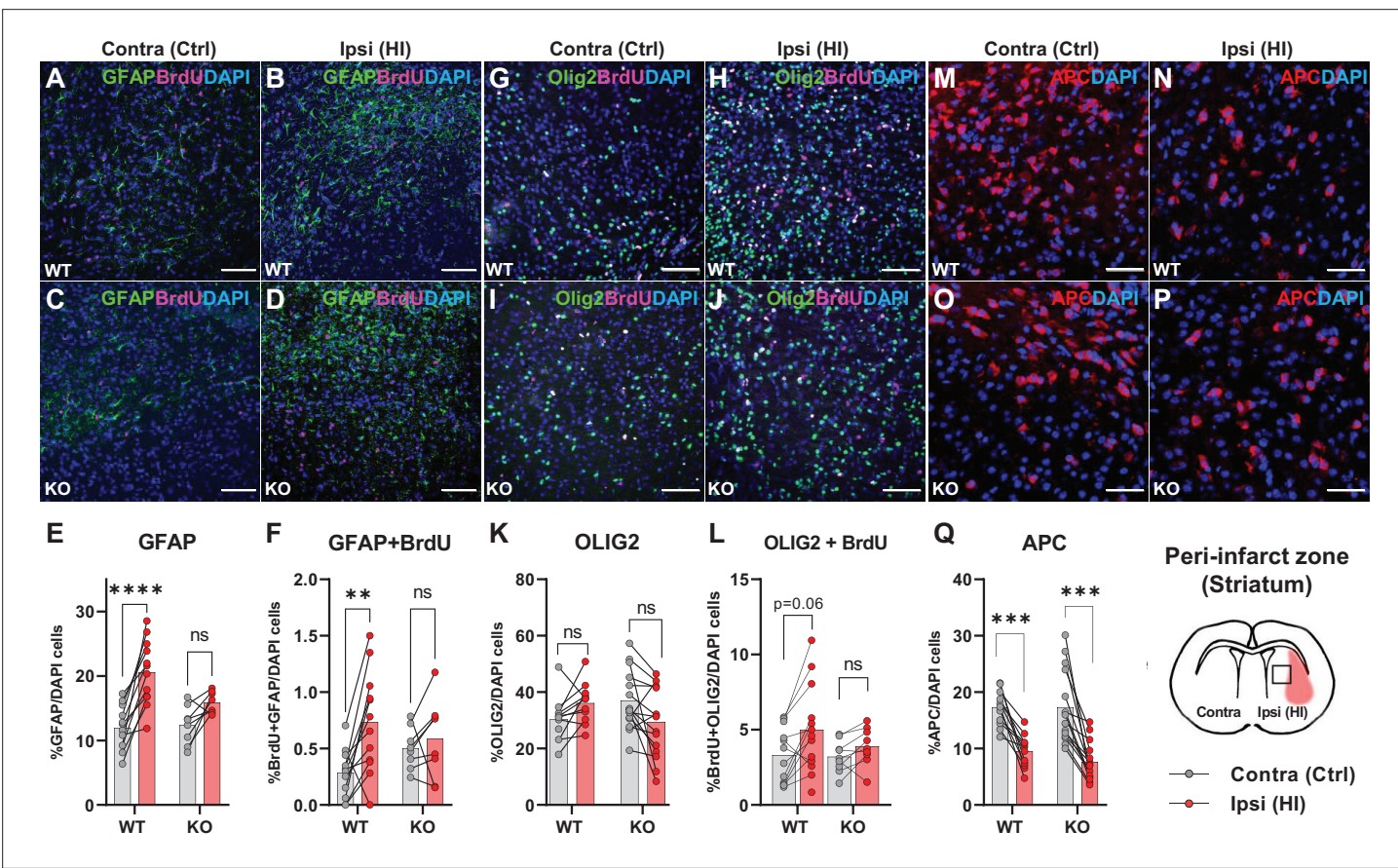

**Figure 4.** Astrocyte and oligodendrocyte proliferation after HI. (**A-D**) Confocal images showing immunolabelling of GFAP +Astrocytes (green) and proliferated BrdU + cells (magenta) in striatal peri-infarct area from contralateral (control, ctrl) and ipsilateral hemisphere (hypoxic ischemia, HI) (see illustration at the bottom right). (**E**) Striatal GFAP + cells. WT contra vs ipsi, p=0.001, df 18, n=12. KO contra vs ipsi, p=0.07, df 18, n=8. (**F**) Proliferated GFAP + cells. WT contra vs ipsi, p=0.002, df 18, n=12. KO contra vs ipsi, p=0.81, df 18, n=8. (**G–J**) Images of striatal Olig2 +oligondendrocytes (whole oligodendrocyte lineage) and BrdU + cells. (**K**) Olig2 +oligodendrocytes, WT contra vs ipsi, not tested due to insignificant one-way ANOVA, df 50, n=12. KO contra vs ipsi, not tested, df 50, n=15. (**L**) Olig2 +and BrdU + cells, WT contra vs ipsi, p=0.06, df 38, n=12. KO contra vs ipsi, p=0.92, df 38, n=9. (**M–Q**) Striatal mature oligodendrocytes (APC). WT contra vs ipsi, p<0.001, df 50, n=12. KO contra vs ipsi, p<0.001, df 50, n=15. Each point represents one sample/mouse. Ipsi- and contralateral samples from the same mouse are indicated with a line. p-Values were calculated with Šídak method for multiple comparisons of selected groups.

The online version of this article includes the following source code and figure supplement(s) for figure 4:

**Source code 1.** Script GFAP analysis.

**Figure supplement 1.** Density measurements of GFAP, Olig2, and APC in the peri-infarct zone.

hemisphere after HI in WT mice. In HCAR1 KO mice on the other hand, there was no significant increase of oligodendrocyte density in the ipsilateral hemisphere (*Figure 4—figure supplement 1C-D*). Thus, investigation of the oligodendrocyte density suggest that HCAR1 may be involved in proliferation of oligodendrocytes as well. The lack of significant increase in Olig2/DAPI ratio in both genotypes after HI demonstrate that the density of other cell types (e.g. DCX +progenitors and microglia) have increased more than the oligodendrocytes.

Finally, by using the marker adenomatous polyposis coli (APC, clone CC1), which labels mature and pre-oligodendrocytes (but not oligodendrocyte precursors), we found that the cell ratio was decreased in the ipsilateral hemisphere after HI in HCAR1 KO as well as WT (*Figure 4M–Q*, %APC+/DAPI: WT contra 17.2±3.1, ipsi 9.5±2.9, p<0.001, KO contra 17.3±6.4, ipsi 7.6±3.4, p<0.001). APC cell density was also decreased after HI in both genotypes (*Figure 4—figure supplement 1E*). This fits with studies showing that ischemia leads to induced apoptosis in mature oligodendrocytes (*Deng et al., 2014*) and arrest of precursor maturation (*Falahati et al., 2013*). The lack of difference between HCAR1 KO and WT suggest that HCAR1 does not initiate oligodendrocyte maturation or inhibit mature oligodendrocyte cell death after HI.

## Weak transcriptional response to HI in the subventricular zone of HCAR1 KO mice

To investigate the mechanisms underlying HCAR1 involvement in brain tissue regeneration, we performed a genome-wide transcriptome analysis by RNA sequencing of the subventricular region from the affected ipsilateral and contralateral (control) hemispheres of mice after cerebral HI. Principal component analysis (PCA, *Figure 5A*) showed that tissue samples from the ipsilateral hemisphere of WT mice clustered away from the contralateral samples (i.e. they showed a different gene expression profile), indicating a strong transcriptional response to HI. Samples from the contralateral hemisphere of HCAR1 KO mice had a comparable gene expression pattern to WT contralateral samples. Notably, ipsilateral HCAR1 KO samples also clustered close together with WT and KO contralateral samples. Thus, it appears that the transcriptional response to HI in the subventricular zone of HCAR1 KO is severely impaired. The number of differentially expressed genes (DEGs) between the different experimental groups further reflected an inadequate response in HCAR1 KO (*Supplementary file 1*): while the WT ipsilateral hemisphere had 7,332 DEGs when compared with WT contralateral hemisphere, indicating a distinct response to HI, only 752 DEGs were detected between HCAR1 KO ipsilateral and contralateral hemispheres. Further, when comparing WT contralateral with HCAR1 KO contralateral hemisphere, only 11 DEGs were identified, whereas WT ipsilateral versus KO ipsilateral identified 6640 DEGs. Therefore, in the undamaged contralateral side, WT and HCAR1 KO showed a similar gene expression profile, while HI induced a large gene expression response in WT that was strongly reduced in HCAR1 KO. To investigate whether the deficient transcriptional response to HI was specific to the subventricular zone, we performed a similar RNA sequencing analysis of the ipsilateral and contralateral hippocampi from the same mice 3 days after HI. In the hippocampal samples, PCA analysis showed a close clustering of HCAR1 KO and WT samples after HI (not shown), indicating a similar transcriptional response to HI. In line with this, we only identified 37 DEGs when comparing HCAR1 KO with WT hippocampi after HI (*Supplementary file 1*). This indicates that HCAR1 acts as a key transcriptional regulator of ischemia response in the subventricular zone but not in the hippocampus.

We then performed gene set enrichment analysis of the subventricular zone samples to identify differentially regulated pathways between the two genotypes in this area. Several pathways were differentially expressed in HCAR1 KO ipsi versus WT ipsi (*Figure 5B*. For extensive maps of differentially regulated pathways between all experimental groups, see *Figure 5—figure supplements 1–4*). Of particular interest to our previous findings, we found the cell cycle pathway strongly downregulated in HCAR1 KO. The relative expression of differentially regulated cell cycle genes across the four different experimental groups is shown in *Figure 5C*. The downregulation of cell cycle genes in HCAR1 KO compared with WT may explain the deficient cell proliferation in HCAR1 KO after HI (*Figures 2–4*).

To validate the transcriptome findings on the protein level, we chose two cyclins that were differentially expressed in the gene set enrichment analysis, namely Cyclin D2 and B1, on which we performed Western blot analysis. The western blots were performed on protein extracts from the ipsilateral striatum (including the SVZ). On average, the Cyclin D2 protein levels in the HCAR1 KO were only 67% of

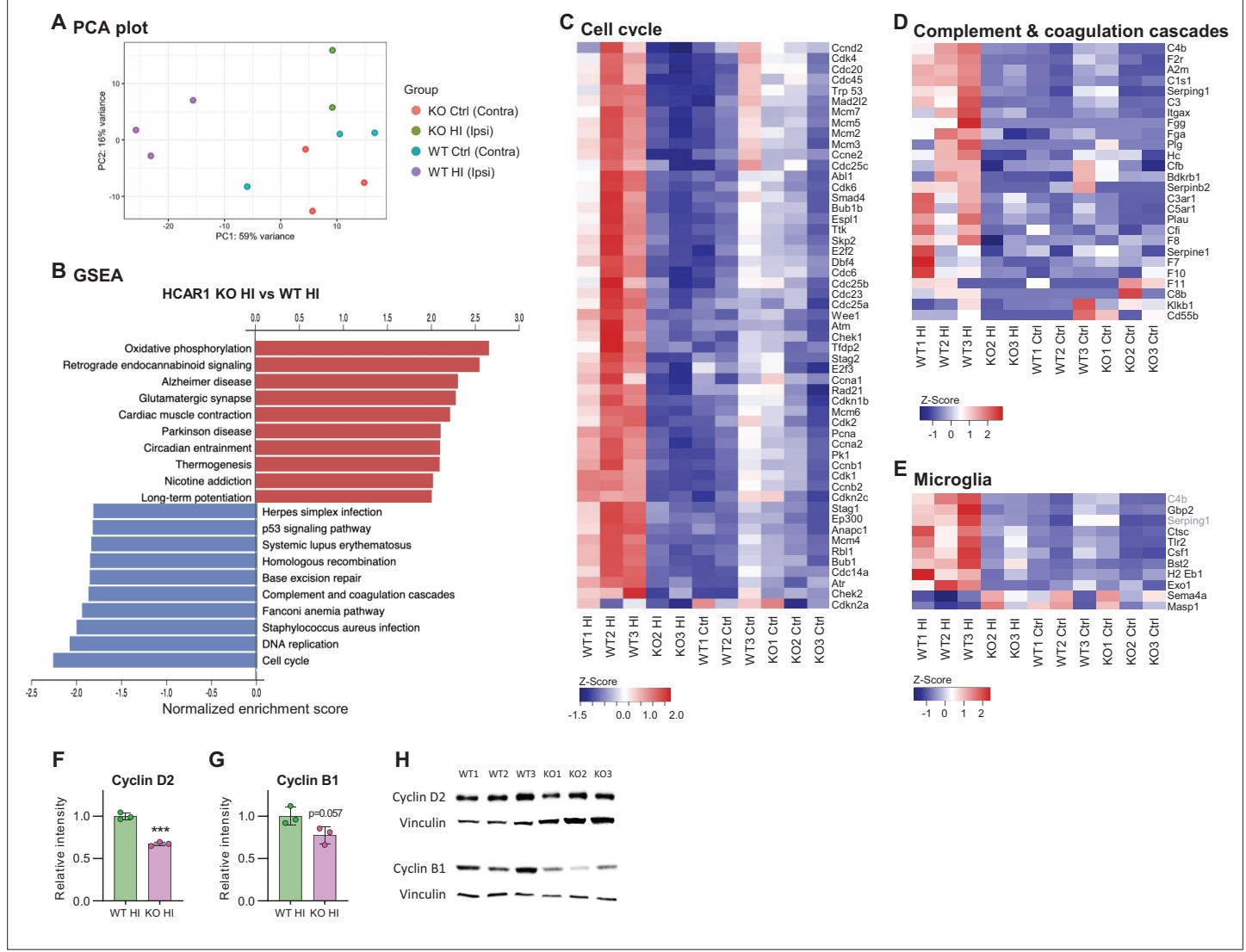

**Figure 5.** HCAR1 regulates transcriptional response to ischemia including cell cycle and complement pathway. (**A**) PCA plot of transcriptome data from subventricular zone tissue samples from the ipsilateral (HI-damaged) and contralateral (control, ctrl) hemisphere in HCAR1 KO and WT mice. Each point represents one sample/mouse. Each colour represents a group. (**B**) Gene set enrichment analysis (GSEA) of HCAR1 KO HI versus WT HI showing the ten most up- or downregulated pathways (FDR <0.05). (**C**) Heatmap showing relative expression of a subset of differentially expressed genes (DEGs) enriched in the cell cycle. (**D**) DEGs related to complement and coagulation cascades (immune system response involved in activation of microglia). (**E**) DEGs that are markers of microglia activation (based on *DePaula-Silva et al., 2019*). Genes shown in (**C–E**) were significantly differentially expressed in HCAR1 KO HI versus WT HI. (**F–H**) Protein expression of Cyclin D2 and Cyclin B1 in the ipsilateral striatum of WT and HCAR1 KO mice after HI. (**F–G**) Graphs presenting relative intensity of striatal Cyclin D2 and B1 from the western blots shown in H. (**F**) Cyclin D2 HCAR1 KO HI vs WT HI, p<0.001, df 4, n=3. (**G**) Cyclin B1 HCAR1 KO HI vs WT HI, p=0.057, df 4, n=3. Data in **F-G** are shown as mean ± SD and p-values were calculated with two-tailed t-test.

The online version of this article includes the following source data and figure supplement(s) for figure 5:

**Source data 1.** Raw images of western blots in *Figure 5*.

**Figure supplement 1.** Gene set enrichment analysis (GSEA) of subventricular zone from HCAR1 KO ipsi versus WT ipsi.

**Figure supplement 2.** Gene set enrichment analysis (GSEA) of subventricular zone from WT ipsi (HI) versus WT contra (control).

**Figure supplement 3.** Gene set enrichment analysis (GSEA) of subventricular zone from HCAR1 KO contra versus WT contra.

**Figure supplement 4.** Gene set enrichment analysis (GSEA) of subventricular zone from HCAR1 KO ipsi (HI) versus HCAR1 KO contra (control).

the WT levels (*Figure 5F and H*. Relative expression %: KO ipsi 67.4±2.4, WT ipsi 100±4.1, p<0.001). The average B1 levels were not statistically different between genotypes (p=0.057), (*Figure 5G–H*. Relative expression %: KO ipsi 77.7±10.1, WT ipsi 100±11, p=0.057). Thus, the reduced transcription of cell cycle genes in HCAR1 KO after HI could to some degree be confirmed on the protein level.

The complement and coagulation cascade pathways were down-regulated in the ipsilateral hemisphere of HCAR1 KO (*Figure 5B and D*) compared with WT mice. This is of particular interest in light of the diminished microglia response in HCAR1 KO as the complement system is involved in microglia activation (*Fumagalli et al., 2015*; *Hammad et al., 2018*). A high number of markers for activated microglia were also down-regulated in HCAR1 KO vs WT after HI (*Figure 5E*), also in line with the impaired microglia activation observed by immunostaining (*Figure 3*). In sum, the subventricular zones of HCAR1 KO mice display a strongly impaired transcriptional response to HI. This can explain the impaired cell proliferation and microglia activation after HI, suggesting that HCAR1 is a key transcriptional regulator of tissue repair after ischemia.

## Discussion

We report that HCAR1 KO mice have a substantial deficit in the restoration of brain tissue after neonatal HI, indicating that lactate receptor HCAR1 plays a crucial role in the processes that lead to tissue repair. Since no exogenous lactate was administered in our experiments, the observed effect must be due to endogenous lactate or possible baseline receptor activity. Indeed, a similar HI mouse model showed a lactate rise in the ipsilateral hemisphere (*Mikrogeorgiou et al., 2020*). Moreover, the lactate level rises in neonatal humans and piglets after a hypoxic-ischemic episode (*Roelants-Van Rijn et al., 2001*; *Wu et al., 2018a*; *Zheng and Wang, 2017*). It is likely that administration of lactate could further leverage the effect of HCAR1: two recent studies showed that mouse pups injected with lactate before, or in the hours or days following HI had improved recovery (*Roumes et al., 2021*; *Tassinari et al., 2020*). The authors suggested that the protective effect of lactate was mainly due to lactate being used as a metabolite to make ATP (*Roumes et al., 2021*; *Tassinari et al., 2020*). However, our data suggest that recovery after lactate injection is partly mediated by HCAR1, which promotes a stronger transcriptional response to HI and thereby facilitates neurogenesis and tissue regeneration after injury. On the other hand, these studies also showed an effect of lactate injections on acute infarct volume. Therefore, a putative explanation is that lactate injected before or immediately after HI reduces lesion size by mainly working as a metabolite, whereas lactate injected at a later time point in large works via HCAR1. Lactate injections can also be protective after ischemic stroke in adult mice (*Berthet et al., 2009*; *Berthet et al., 2012*; *Buscemi et al., 2020*; *Castillo et al., 2015*). Here, it also seems to be a combination of metabolic and HCAR1-dependent effects since replacing lactate with either the HCAR1 receptor agonist 3,5-dihydroxybenzoic acid (3, 5 DHBA) or the metabolic substrate pyruvate offered partial protection (*Castillo et al., 2015*). In the current study, we induced a permanent occlusion of the left carotid artery. In human HI, reperfusion often occurs, which can lead to reperfusion injury. Although lactate administrations has been shown to reduce reperfusion injury in adult mice and humans (*Annoni et al., 2021*), a recent study from adult rats suggested that the effect was HCAR1 independent (*Buscemi et al., 2021*). We are not aware of any studies on the effect of lactate and HCAR1 on reperfusion injury in neonatal HI.

An ischemic event will induce a significant transcriptional response in the neonatal as well as the adult brain (*Androvic et al., 2020*; *Hedtjärn et al., 2004*). By RNA sequencing of tissue samples from the subventricular zone, we found that HCAR1 KO mice displayed a weak transcriptional response to HI, with a 90% reduction in DEGs compared with WT mice. Hence, HCAR1 appears to be important for the induction of the transcriptional response to HI. This HCAR1 dependence is specific for the subventricular zone, as hippocampal samples showed little difference between WT and HCAR1 KO mice in the transcriptional profile after HI. In line with our findings, a recent study shows an effect of HCAR1 activation on neurogenesis in the subventricular zone and not in the hippocampal subgranular zone (*Lambertus et al., 2020*). Furthermore, the structure and cellular composition of the two neurogenic niches are in themselves unique and react differently to HI in the neonatal and adult brain (*Semple et al., 2013*). A possible role of HCAR1 in other, more recently discovered neurogenic areas (*Li et al., 2009*) remains to be investigated.

An essential part of the repair process after a neonatal brain injury is the generation of new cells. This occurs by increased proliferation and differentiation of progenitor cells and involves upregulation

of genes involved in the cell cycle pathway (*Prasad et al., 2012*; *Wen et al., 2005*). By use of neurosphere assays, we showed that neural progenitor cells from HCAR1 KO mice have reduced proliferation ability. Moreover, while WT mice doubled the ratio of proliferating cells after HI in the striatal SVZ-IZ, HCAR1 KO mice were unable to increase cell proliferation (*Figure 2P*). In the V-SVZ, there was no difference in cell proliferation post ischemia in the WT or HCAR1 KO mice. This lacking proliferative response at the basal part of the SVZ has been reported earlier in a study using the same HI-protocol as we present here (*Sejersted et al., 2011*). The same study showed similar reactive neurogenesis in the deeper part of the SVZ and IZ in WT mice. Importantly, the SVZ is in a transitional phase at the time of our analyses: during the first weeks of development, the SVZ shrinks from being up to 300 μm thick, to under 100 μm, and shifts closer to the VZ (*Fiorelli et al., 2015*). In the same period, there is a gradual decrease in the embryonic radial neuronal migration and a shift to the adult rostral neuronal migration towards the olfactory bulb (*Inta et al., 2008*). Upon a hypoxic ischemic insult, the radial pathway reopens and provides striatum and cortex with cells from the SVZ in adult (*Goings et al., 2004*) and postnatal/young mice (*Covey et al., 2010*). Therefore, the increased number of progenitors we observed in the SVZ-IZ, presumably represent either radial embryonic or reactive neuronal migration/neurogenesis. Given that we did not see any differences between the genotypes in the number of progenitors in the contralateral SVZ-IZ (*Figure 2*), we conclude that the increased progenitors at the ipsilateral hemisphere represent HI-reactive neurogenesis.

Despite the lack of differences in cell proliferation between the contralateral WT and HCAR1 KO in vivo, the in vitro cultured neurospheres showed reduced proliferation in HCAR1 KO compared with WT. This discrepancy could be due to compensatory mechanisms in vivo: as neurospheres lack vasculature and the extracellular milieu that is present in vivo, they may not be able to compensate a lack of HCAR1-driven proliferation. It is also possible that the experimental steps needed to produce neurospheres cause a stress response in the cells that does not occur in the contralateral hemispheres in vivo.

In addition to the effect of HCAR1 on neurogenesis, our analyses of astrocytes, microglia and oligodendrocyte suggest that HCAR1 can affect proliferation post HI in glial cells. Our analysis was done as density of cells per area (in Supplements to *Figures 2–4*) and as ratio per total number of cells (in *Figures 2–4*). While the cell densities per area give information about the overall change in each cell population, the cell ratios say how much a cell population has changed in comparison to the total density of cells. A significant increase in cell ratio for a cell type would require that this cell type increased more than the total cell increase. In WT mice, this was the case for proliferating microglia and astrocytes, but not for oligodendrocytes, indicating that the density of proliferating microglia, astrocytes and neuronal progenitors had increased more than that of oligodendrocytes. Still, the density of proliferating oligodendrocytes per se was increased after HI in WT mice, but not in KO mice. Therefore, we argue that HCAR1 KO affects proliferation after HI in all the analysed cell lineages.

Transcriptome analysis of the subventricular niche revealed that the cell cycle genes were strongly upregulated after HI in WT, but not in HCAR1 KO mice. Hence, it seems that HCAR1 can act as a transcriptional regulator of cell cycle genes, thereby controlling cell proliferation. It is important to note that transcriptome data do not always correspond with proteome data, and therefore a proteome analysis would be useful to confirm the transcriptome results in this study. Nevertheless, the combination of transcriptome data with western blots partly confirming down-regulation of some cell cycle genes, and the immunostainings showing reduced cell proliferation after HI, support our hypothesis. A role of HCAR1 in cell proliferation was previously shown in cancer and osteoblast cell lines (*Stäubert et al., 2015*; *Wu et al., 2018a*). Altogether, these findings may provide new insights of importance for treatments of hypoxic ischemic brain injury in the perinatal period and perhaps in the adult brain. However, induced reactive proliferation is increased in the postnatal mice (P9) compared with adult mice, and the time window of this robust induced neurogenesis decreases substantially within the next two weeks of the rodents life (*Covey et al., 2010*).

Microglia carry out several vital functions in response to brain injury. These include clearance of damaged tissue, resistance to infections and restoration of tissue homeostasis (*Jin and Yamashita, 2016*). We detected an increase in the proliferation and activation of microglia in the peri-infarct zone after HI in WT, but not in HCAR1 KO mice. The lack of microglia proliferation could be due to the reduced induction of cell cycle response, as discussed above. The absence of microglia activation was

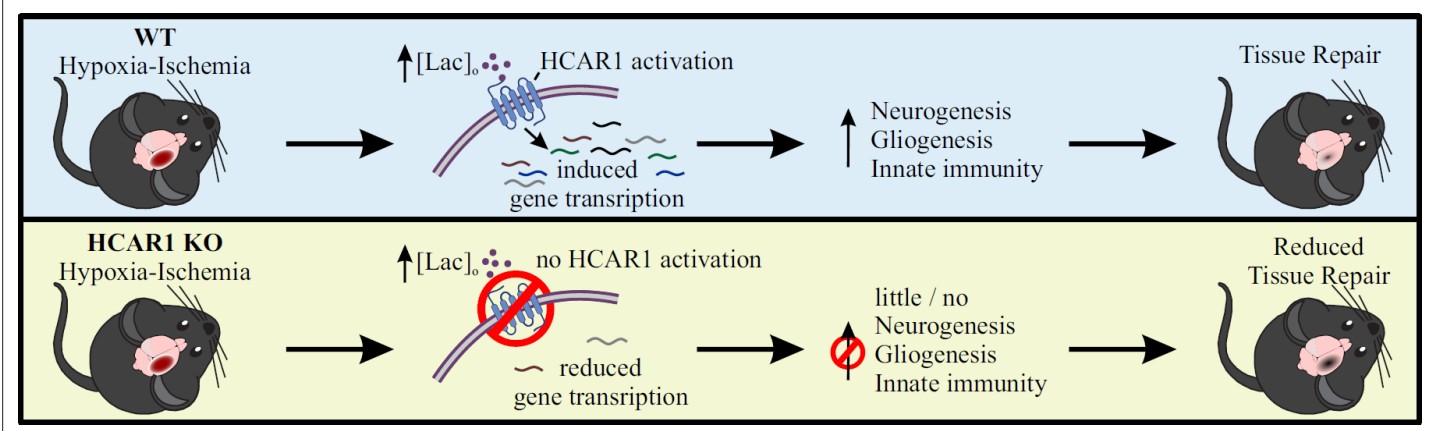

**Figure 6.** Proposed model for the role of HCAR1 in neonatal HI. During and after HI, the extracellular concentration of lactate ($[lac]_o$) is elevated. **Top panel:** In WT mice, the elevated lactate causes HCAR1 activation, which induces transcription of genes involved in tissue response to ischemia. This includes genes responsible for neurogenesis, proliferation of glial cells and innate immunity, thereby promoting tissue repair. **Bottom panel:** In HCAR1 KO mice, the transcriptional response to ischemia is severely reduced. Without the HCAR1-induced gene transcription, there is little cell proliferation and innate immune response, which in turn gives an impaired tissue repair.

confirmed on a transcriptional level, as genes associated with activated microglia were upregulated in WT but not in KO. This effect may be explained by the reduced complement system response in HCAR1 KO. Altogether, the differences in microglial data suggest that the damaging effects of HCAR1 KO in HI are due to interplay of both the immune response and proliferation and regeneration of the damaged brain cells.

In addition to the effects on cell cycle and microglia activation discussed above, the transcriptome analysis revealed a large number of differentially expressed genes and pathways, including genes involved in DNA repair and glutamate signalling. These processes will likely also influence the ability of the brain tissue to repair after injury.

Apoptotic pathways are upregulated after HI and are a major contributor to cell loss. But our transcriptome data did not reveal any differences between the genotypes in expression of apoptotic genes. There was also no significant upregulation of apoptotic pathways after HI within each genotype. Our RNA seq was performed on SVZ tissue, which lies a distance away from the infarcted area and may therefore experience less HI-induced apoptosis. However, apoptosis was also not upregulated in the hippocampus after HI (data not shown). Another possible explanation for the lack of apoptosis in our RNA seq data could be the timing: apoptotic activity has been shown to peak around 24 hr after HI, at least in rats (*Wang et al., 2001*). Still, we cannot exclude the possibility that a transcriptome analysis from an area closer to the infarct core (e.g. the striatal peri-infarct zone), and potentially taken at a different time point after HI, would have revealed differences in apoptotic pathways between HCAR1 KO and WT. Thus, a potential difference in the number of dead cells could also contribute to the different outcomes in the two genotypes.

Overall, our data show that HCAR1 is a key transcriptional regulator of brain tissue response to an ischemic insult. We therefore propose a model in which activation of HCAR1 by elevated lactate during and after HI stimulates a transcriptional response involving pathways responsible for tissue repair (*Figure 6*). HCAR1 could be a target of future treatment for neonatal HI and possibly other forms of brain injury.

## Materials and methods

### Animals

HCAR1 KO and C57Bl/6 N (WT) mice were used for this study. The HCAR1 KO line was a gift from Prof. Dr Stefan Offermanns, Max Planck Institute for Heart and Lung Research, Bad Nauheim, Germany and has been described previously (*Ahmed et al., 2010*). The KO line was maintained in C57Bl/6 N background, and genotypes were determined by PCR analysis with DNA samples extracted from mouse ears. All mice were housed in a climate-controlled environment on a 12 hr light/dark cycle with

free access to rodent food and water. All efforts were made to reduce the number of animals used in experiments. Both females and males were included in the analyses. The mice were treated in accordance with the national and regional ethical guidelines and the European Union's Directive 86/609/EEC. Experiments were performed by FELASA-certified personnel and approved by the Norwegian Animal Research Authority.

## Mouse model for cerebral HI

Cerebral hypoxia and ischemia (HI) was produced in P9 mice by permanent occlusion of the left common carotid artery (CCA) followed by systemic hypoxia, as previously described (*Sejersted et al., 2011*). In brief, pups were anaesthetised with isoflurane (4% induction in chamber, 2.5% maintenance on mask in a 2:1 mixture of ambient air and oxygen), and a skin incision was made in the anterior midline of the neck. The left CCA was exposed by blunt dissection and carefully separated from adjacent tissue. A needle was placed the CAA, and a monopolar cauteriser (Hyfrecator 2000; ConMed) at a power setting of 4.0 W was used to electrocoagulate the artery. The neck incision was closed with absorbable sutures (Safil 8–0 DRM6; B. Braun Melsungen AG). The surgical procedure was completed within 5 min. After a recovery period of 1–2 hr, the pups were exposed to a hypoxic (10% oxygen balance nitrogen; Yara), humidified atmosphere for 60 min at 36.0 °C. The pups were returned to their dam and after 6 hr, 24 hr or 42 days brains were retrieved and prepared for immunohistochemistry, cell culture experiments, or RNA sequencing.

## Measurement of acute tissue damage and long-term tissue loss

Mice were terminated by neck dislocation 24 hr or 42 days after HI. Brains were removed from the skull and freed from dura mater and vascular tissue before being transferred to a precooled brain mould immersed in ice-cold PBS. One mm coronal slices were cut using an adult brain slicer (51–4984; Zivic Instruments, Pittsburgh, PA, USA). For measurement of acute tissue damage, sections were soaked in 2% TTC (T8877, Sigma) in PBS for 30 min at room temperature and subsequently fixed in 4% paraformaldehyde (PFA, Sigma-Aldrich, St. Louis, MO, USA; 15,812–7) in PBS at 4 °C for 1 hr. Photos were captured with a digital camera (Nikon D80), and pictures were analysed using image J software (NIH, San Francisco, CA, USA). Quantification of acute tissue damage/infarct size was carried out as previously described (*Sejersted et al., 2011*). Briefly, the infarct area was calculated by subtracting the area of undamaged, TTC positive tissue in the ipsilateral hemisphere from that of the intact contralateral hemisphere, thereby correcting for brain oedema. The relative size of the damage was expressed as per cent of the contralateral hemisphere. Volume loss was calculated by modified Cavalieri's principle, using the formula $V=\sum APt$ where V is the total volume, $\sum A$ is the sum of the areas measured, P is the inverse of the section sampling fraction and t is the section thickness. For measurement of long-term tissue loss 42 days after HI, coronal sections were prepared as described above, but without TTC staining. Sections were fixed in 4% paraformaldehyde (PFA) in PBS for 30 min, and 10% formalin for 24 hr and photos were taken. Tissue loss was calculated by subtracting the total volume or section volume of the ipsilateral hemisphere from that of the contralateral hemisphere. One brain (HCAR1 KO) was excluded from the analysis due to so excessive damage that the sections fell apart, thus making measurements difficult. The person performing measurements was blinded to genotype during the measurements.

## Neurosphere cultures and assays

Neurospheres derived from forebrains of C57BL/6 and C57BL/6 HCAR1 knockout mice at postnatal day 3 were generated and propagated as previously described with modifications (*Wang et al., 2010*). Briefly, dissected brain tissues were finely minced and cultured with proliferation medium of DMEM/F12 (Invitrogen) supplemented with 2 mM glutaMax, 20 ng/ml EGF (R&D Systems), 10 ng/ml bFGF (R&D Systems), N2 supplement (ThermoFisher Scientific), B27 supplement without vitamin A (ThermoFisher Scientific), and penicillin/streptomycin. Under the proliferating condition, cells were grown as free-floating neurospheres. After 7 days in culture, cells in primary neurospheres were trypsinised with trypsin-EDTA (Invitrogen), dissociated mechanically, and placed onto 75 cm2 flasks and the neurospheres were passaged every 7 days. For neurosphere self-renewal, dissociated single NSPCs were plated at a density of $1.0 \times 10^4$ per well onto 6-well suspension plates with proliferation medium. After 10 days in culture, images of the entire well were captured with EVOS microscope.

The pictures were analysed using the ImageJ software to obtain the total number and average size of neurospheres per well. For the neurosphere differentiation, dissociated single NSPCs were plated at a density of $5 \times 10^4$ cells per centimetre square onto tissue culture plates pre-coated with poly-D-lysine (Sigma). Cells were cultured in differentiation medium (proliferation medium minus EGF and bFGF) for 5 days with half medium changes daily, and the cells were fixed at different time points for further experiments.

## Immunocytochemistry and quantification of neurospheres

Immunocytochemistry was performed as described previously (*Wang et al., 2010*). Differentiated NSPCs were fixed with 4% paraformaldehyde and treated with 0.1% Triton X-100/PBS. Following blocking with 5% BSA, 5% goat serum, and 0.1 Triton X-100 in PBS for 30 min, the cells were incubated with monoclonal anti-neuron-specific beta-III tubulin (Tuj-1, MAB1195 R&D Systems), rabbit polyclonal astrocyte-specific anti-glial fibrillary acidic protein (GFAP, Z0334 Dako) in PBS containing 0.5% BSA, 0.5% goat serum, and 0.1% Tween 20 at 4 °C overnight. Then the cells were incubated with fluorescent anti-mouse or rabbit secondary antibody (Alexa 594 and Alexa 488, Molecular Probes). The nuclear dye 4',6-diamidino-2-phenylindole (DAPI) at 1 l ng/ml (Molecular Probes) was added to visualise all cells. To obtain the percentage of each cell type, 4000–5000 cells that were morphologically identified in 10 random fields from two different cultures were counted under a 10 x objective. Percentage of positive cells was calculated in relation to the total number of cells, as detected by DAPI nuclear staining.

## BrdU incorporation in mice

For the BrdU experiments, we injected 0.1 mg/g of BrdU into the peritoneum at day 4–7 after hypoxic ischemia at 24 hr intervals. Animals were transcardially perfused 2 hr after the last injection and brains were fixed and stained as described below.

## Preparation of mouse tissue and immunolabelling

Mice were anesthetised and transcardially perfused with 4% PFA. Brains were then removed and stored in 4% PFA for 24 hr, and then immersion fixed in 10% formalin until paraffin embedding. We then cut 6- to 8-µm-thick coronal sections through the entire forebrain using a microtome (Thermo-Scientific, Waltham, MA, USA). For immunostaining, sections were heated in an incubator at 60 °C for 30 min. Deparaffinisation/dehydration was performed by immersing in Neoclear (2 × 5 min, Millipore, Darmstadt, Germany) followed by rehydration in an EtOH gradient (100% 2 × 5 min; 96% 5 min and 70% 5 min) and then transferred to MQ $H_2O$. Sections were then incubated at 100 °C in citrate antigen retrieval buffer (pH 6.0) for 20 min using a coverslip-paperclip method described by *Vinod et al., 2016*. Following antigen retrieval, slides were incubated in blocking solution (10% normal goat serum, 1% bovine serum albumin, 0.5% Triton X-100 in PBS) for 1 hr. Primary antibody incubation was done at room temperature overnight. Primary and secondary antibodies were diluted in a solution containing 3% normal goat serum, 1% bovine serum albumin, 0.5% Triton X-100 in PBS. The next day, sections were washed 3 × 10 min in PBS and then incubated with secondary antibodies for 1 hr at room temperature before a new wash of 2 × 10 min in PBS and a third incubation with DAPI for 15 min. Sections were rewashed (3 × 10 min) before being mounted with ProLongTM Glass Antifade Mountant (Fisher Scientific, Waltham, Massachusetts). Cover glass thickness was 0.13–0.17 mm. Primary antibodies were rat anti-BrdU (AB6326 Abcam, 1:200), guinea pig anti-Doublecortin (AB2253 Abcam, 1:500), mouse anti-GFAP (Sigma G3893) and rabbit anti-Iba1 (019–19741 Wako, 1:500). Secondary antibodies used were Alexa 488 goat anti-rabbit, Alexa 555 goat anti-rat, Alexa goat anti-guinea pig (all diluted 1:400).

## Analysis of immunolabelling

Images were captured on a Leica SP8 confocal microscope, using a 20 x objective (n.a. 0.75, microglia, astrocyte and oligodendrocyte experiments) and a 40 x oil-immersion objective (n.a. 1.3, DCX experiments). All image analysis was done in Fiji ImageJ. Before analysing, Z-stacks were flattened with maximum z-projection. For the doublecortin analysis (*Figure 2*), we analysed four images per hemisphere from two different sections. The analysis of the subventricular niche was divided in two distinct areas, were we defined the ventricular-subventricular zone at 0–50 µm from the ependyma and the

subventricular-intermediate zone as 50–200 µm. We counted cells manually using the Cell Counter plugin in Fiji Image J with operators blinded during imaging and analysis. In the microglial experiments, the blinded operators defined the ischemic core and peri-infarct zone by the morphological appearance of microglia, cell-cores and general cyto-architectural integrity. In the microglia analysis (*Figure 3*), we used the WEKA-segmentation computer learning algorithm (*Arganda-Carreras et al., 2017*) for image segmentation. After computer training, the images were automatically segmented using the trained algorithms (Source data files 2-4). The astrocyte and oligodendrocyte images were segmented using Otsu threshold algorithm (*Figure 4*). After segmentation cells were analysed automatically utilising the Analyze Particle tool with scripts written in ImageJ (Source data file 1). In the microglia experiment, we analysed two images per hemisphere from two different sections. For microglia counting, we counted Iba1 +overlapping with DAPI +objects, and then triple overlap with BrdU for counting of newly made microglia. Based on previous literature on microglia morphology during cerebral ischemia (*Morrison and Filosa, 2013*), we selected the average maximum branch length and microglial cross-sectional size as data for activated microglia morphology analyses. For the branch and size analyses, we excluded processes protruding from out of focus microglia by only analysing Iba1 +objects above 95 $\mu m^2$. Branches were analysed using the skeletonise (2D/3D) and analyse skeleton (2D/3D) functions in ImageJ.

## RNA sequencing and analysis

Subventricular zone (SVZ) and hippocampal tissue was dissected from the ipsilateral (damaged) and contralateral (undamaged) hemisphere of mice 3 days after HI. For SVZ dissection, a razor blade was used to dissect out a 2-mm-thick coronal section containing the rostral and middle part (main body) of the lateral ventricles. The two hemispheres were then separated. Under a dissection microscope, a sharp spatula was used to scrape off a thin layer of the SVZ from below the corpus callosum and down to the bottom (ventral edge) of the lateral ventricle. The samples were snap-frozen in liquid $N_2$ and stored at –80 °C before RNA isolation. RNA isolation was performed using QIAGEN allprep kit, and final RNA was dissolved in MQ $H_2O$ and stored at –80 °C. Paired-end sequencing was performed with the Illumina platform by BGI Tech Solutions (Hong Kong). The quality control of fastq files was performed with FastQC v0.11.9 (*Andrews, 2022*). Alignment to reference genome (GRCm38) was accomplished with hisat2 v2.1.0 (*Kim et al., 2015*) while annotation and count matrix was completed with featureCounts v.2.0.0 (*Liao et al., 2014*). We performed downstream DEGs analysis in R v3.6.1 with DESeq2 v1.24.0 (*Love et al., 2014*). One HCAR1 KO1 ipsi sample was removed from further analysis as it did not cluster with any of the other samples and therefore appeared as an outlier (not shown). GSEA (Gene Set Enrichment Analysis) was done with WebGestalt (*Liao et al., 2019*). Heat maps were generated in R v3.6.1 with heatmap3 v1.1.7 (*Zhao et al., 2014*).

## Western blot

Mice were sacrificed 3 days post HI, and the striatum was isolated. Then, 60 µL RIPA buffer (300 mM NaCl, 50 mM Tris pH7.5, 1 mM EDTA, 0.1% SDS, 0.5% Sodium deoxycholate, 0.1% Triton X-100) supplemented with protein inhibitor cocktail and 20 nM dithiothreitol were added to the isolated striatum samples before sonication. Sonication was performed at 20–30% amplitude for 7–20 s depending on tissue size. Following sonication, the samples were kept on ice for 15 min before centrifugation at 4°C for 15 min. The supernatant was collected, and protein concentration determined, then mixed with 4 X NuPAGE LDS sample buffer (life technologies, #NP0008) and heated at 70°C for 10 min. 5–20 µg protein were loaded and separated by 4–12% SDS page (Invitrogen, #NW04125BOX), and then transferred to a 0.2 µm PVDF membrane (Bio-rad, #1704156) with the bio-rad Trans-BlotTurbo Transfer system. Following blotting, the membranes were incubated in blocking solution (5% milk in PBS-T) for 1 hr. Primary antibody incubation was done at 4°C overnight. Primary antibodies were diluted in the blocking solution. Following primary incubation, the membranes were washed 3 × 10 min in PBS-T and then incubated in secondary antibodies for 1 hr at room temperature before a second wash of 3 × 10 min in PBS-T. Secondary antibodies were diluted in PBS-T. Protein levels were detected using Supersignal West Femto Maximum sensitivity substrate (Thermo scientific, #34095) and visualized with Bio-rad ChemiDoc MP system. We normalized each band to our loading control Vinculin, we further normalized each value to the average of WT per gel, the values were then averaged from two technical replicates for the final analysis. A total of three biological replicates were

analysed per genotype. We further validated Vinculin as a loading control comparing it to actin to ensure there was no apparent effect of hypoxic-ischemia (data not shown). Primary antibodies were mouse anti-Cyclin B1 (1:500; Abcam, #ab72), rabbit anti-Cyclin D2 (1:1000; Cell Signaling Technology, #37413), and mouse anti-Vinculin (1:50000; Merck life sciences, #V9131). Secondary antibodies used were HRP-labeled donkey anti-mouse (1:10 000; Abcam, #ab6820), and HRP-labeled goat anti-rabbit (1:20 000; EpiGentek, #C10018-1).

## Statistical analysis

P-values in *Figure 1*, *Figure 2G–I* and *Figure 5F–G* are from unpaired, two-tailed, t-test's. In *Figure 2O–R*, *Figure 3* and *Figure 4* we used the Šídak (Šídak-Bonferroni) (*Sidak, 1967*) method for multiple comparisons of selected groups (WT-contra vs KO-contra, WT-ipsi vs KO-ipsi, WT-contra vs WT-ipsi, and KO-contra vs KO-ipsi), the Šídak analysis was only performed if the one-way ANOVA analysis was significant. Degrees of freedom are written as df in the figure legends. All error bars represent the standard deviation. In *Figure 2O–R*, *Figure 3* and *Figure 4* there are no error-bars as all the individual data points are shown in the graphs. No formal power analyses were used to predetermine sample size. All experimental units were included in the analyses (none were excluded), unless otherwise stated.

## Stem- and progenitor cells and terminology

There are different types of neural stem- and progenitor cells, and different terminology exists. Our in vivo data present SVZ-derived type A Neuroblasts (progenitors). Before differentiation, the neurospheres consist mainly of neural stem-/ progenitor cells. For simplicity, these cells are termed progenitor cells throughout the manuscript.

## Acknowledgements

We thank Prof. Stefan Offermanns and collaborators at the Max-Planck-Institute for Heart and Lung Research, Bad Nauheim, Germany, for providing breeder HCAR1 knockout mice, Dr. Gunn Anette Hildrestrand for assistance with hypoxic-ischemic experiments and Dr. Adam Filipczyk for comments on the manuscript. Images were obtained with support from Drs. Anna Lång and Stig-Ove Bøe at the South-Eastern Health Authority Core Facility of Advanced Light Microscopy (Gaustad, Norway). This work was supported by the South-Eastern Norway Regional Health Authority (grants 2020042 and 2018050), the Norwegian Health Association (grant 4841), the Civitan Alzheimer Fund and the Medical research program at the University of Oslo.

## Additional information

### Competing interests

Linda H Bergersen: LHB is a stock owner and head of the board at Lactate AS. The other authors declare that no competing interests exist.

### Funding

| Funder | Grant reference number | Author |
|---|---|---|
| Helse Sør-Øst RHF | PhD Fellowship 2020042 | Emilie R Glesaaen |
| Helse Sør-Øst RHF | Career Grant 2018050 | Niklas Meyer<br>Johanne E Rinholm |
| Nasjonalforeningen for Folkehelsen | Postdoctoral Fellowship 4841 | Johanne E Rinholm |
| Civitan Norway | Running costs | Johanne E Rinholm |

The funders had no role in study design, data collection and interpretation, or the decision to submit the work for publication.

## Author contributions
Lauritz Kennedy, Conceptualization, Data curation, Formal analysis, Investigation, Resources, Software, Validation, Visualization, Writing – original draft, Writing – review and editing; Emilie R Glesaaen, Data curation, Formal analysis, Investigation, Validation, Visualization, Writing – original draft, Writing – review and editing; Vuk Palibrk, Formal analysis, Investigation, Visualization, Writing – review and editing; Marco Pannone, Formal analysis, Investigation, Software, Visualization, Writing – review and editing; Wei Wang, Formal analysis, Investigation, Validation, Visualization, Writing – review and editing; Ali Al-Jabri, Niklas Meyer, Marie Landa Austbø, Formal analysis, Investigation, Writing – review and editing; Rajikala Suganthan, Formal analysis, Investigation, Methodology, Writing – review and editing; Xiaolin Lin, Formal analysis, Investigation, Supervision, Writing – review and editing; Linda H Bergersen, Conceptualization, Resources, Validation, Writing – review and editing; Magnar Bjørås, Conceptualization, Funding acquisition, Methodology, Supervision, Writing – original draft, Writing – review and editing; Johanne E Rinholm, Conceptualization, Funding acquisition, Investigation, Project administration, Supervision, Validation, Writing – original draft, Writing – review and editing

## Author ORCIDs
Niklas Meyer http://orcid.org/0000-0002-8377-097X
Johanne E Rinholm http://orcid.org/0000-0003-3741-850X

## Ethics
The mice were treated in strict accordance with the national and regional ethical guidelines and the European Union's Directive 86/609/EEC. Experiments were performed by FELASA-certified personnel and approved by the Norwegian Animal Research Authority.

## Decision letter and Author response
Decision letter https://doi.org/10.7554/eLife.76451.sa1
Author response https://doi.org/10.7554/eLife.76451.sa2

# Additional files

## Supplementary files
• Supplementary file 1. Number of differentially expressed genes (DEGs) between the different experimental groups. Thresholds ->logFC ≥ 0.5 (UP DEGs) and ≤–0.5 (DOWN DEGs); all of them significant at adjusted $P$-value <0.05. SVZ, subventricular zone; hc, hippocampus.

• Transparent reporting form

## Data availability
The RNA sequence data are available at Dryad, https://doi.org/10.5061/dryad.8w9ghx3kw. Script for analysis of immunostaining and algorithms used for WEKA segmentation for Figures 3 and 4 are provided in Figure 3—source code 1 - BrdU WEKA, Figure 3—source code 2 - DAPI WEKA, Figure 3—source code 3 - IBA1 WEKA, Figure 3—source code 4 - Script Microglia analysis, Figure 4—source code 1 - Script GFAP analysis. Raw images of western blots in Figure 5 are provided in Figure 5—source data 1.

The following dataset was generated:

| Author(s) | Year | Dataset title | Dataset URL | Database and Identifier |
|---|---|---|---|---|
| Rinholm JE, Kennedy L, Glesaaen E, Palibrk V, Pannone M, Wang W, Al-Jabri A, Suganthan R, Meyer N, Lin X, Bergersen L, Bjørås M | 2021 | HCAR1 KO and WT hypoxia-ischemia RNA sequencing data | https://doi.org/10.5061/dryad.8w9ghx3kw | Dryad Digital Repository, 10.5061/dryad.8w9ghx3kw |

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
