## [Editor Report]

This manuscript highlights the contribution of lactate receptor HCAR1 to mechanisms of neuronal repair after hypoxic injury. This paper will be of interest to scientists studying mechanisms of hypoxia-ischemia-induced brain injury and tissue repair and has high translational relevance. It implicates a novel mechanism, lactate-HCAR1 signaling, as underlying cellular proliferation and neurogenesis needed for tissue regeneration after injury. A series of compelling experiments presented in this article support the necessary role of HCAR1 in these effects.

---

## [Decision Letter]

**Decision letter after peer review:**

[Editors’ note: the authors submitted for reconsideration following the decision after peer review. What follows is the decision letter after the first round of review.]

Thank you for submitting your work entitled "Lactate receptor HCAR1 regulates neurogenesis and microglia activation after neonatal hypoxia-ischemia" for consideration by *eLife*. Your article has been reviewed by 3 peer reviewers, one of whom is a member of our Board of Reviewing Editors, and the evaluation has been overseen by a Senior Editor. The following individual involved in review of your submission has agreed to reveal their identity: Achira Roy (Reviewer #2).

We are sorry to say that, after consultation with the reviewers, we have decided that your work will not be considered further for publication by *eLife*, given the number of items that need to be addressed. Given the general interest and compelling initial findings from this study, the reviewers provide constructive feedback on data interpretation, methods, and specific experiments (specifically rescue experiments) to further support the role of HCAR1 and bring this paper to a level for publication in *eLife*.

*Reviewer #1 (Recommendations for the authors):*

This is a very interesting study implicating HCAR1, a G-protein coupled lactate receptor, in the pathophysiology of hypoxia-ischemia (HI)-induced impairments in brain tissue regeneration, cellular proliferation, neurogenesis and microglial activation. The authors use a HCAR1 KO mouse and provide convincing in vitro and in vivo evidence that newly proliferating cells in the intermediate zone (adjacent to SVZ) ipsilateral to the infarct/HI injury, as well as ipsilateral neurogenesis, is impaired after HI when HCAR1 is not present. Furthermore, microglial proliferation as well as activation is also reduced in the setting of HCAR1 KO. And finally, the authors provide RNA-sequencing data from HI-injured WT and HCAR1 KO mice, demonstrating that the significant differential gene expression occurring after HI in WT mice is not seen in HCAR1 KO mice. Overall, this study would be of major interest to scientists studying mechanisms of HI injury and intervention, possibly informing design of molecular targets to preserve cellular proliferation and tissue regeneration after ischemic injury.

Strengths:

The data presented here are very convincing that HCAR1 is required for tissue regeneration in the chronic period after HI injury (figure1). Figure 2 further shows evidence from both neurospheres and immunostaining that at the level of the intermediate zone, cellular proliferation and differentiation are directly impacted in the absence of HCAR1. Contralateral uninfarcted hemisphere is used as an intra-individual control, and very clearly demonstrates impact on proliferation and neurogenesis is specific to the hemisphere ipsilateral to the damage. It is interesting to note that HCAR1 absence alone does not impact proliferation and differentiation. Figure 3 shows that in the peri-infarct area, microglial activation and proliferation is repressed in HCAR1 KO mice. Figure 4 is descriptive in that it shows cell cycle pathway and complement pathway downregulation in HCAR1 KO SVZ from infarcted side of brain.

Weaknesses:

There is not significance found in the proliferative assessment of the SVZ, in that both WT and HCAR1 mice have reduced proliferation (did not reach significance, however, the figure Supp 1F suggests a strong trend). It is also interesting that there was a transcriptional response to HI in hippocampus in HCAR1 KO but this was not explored further. It would be a logical next step, and one that would enhance the paper, to disclose what the downstream genes discovered from the RNA-seq data set that are involved in neurogenesis and innate immunity that are not activated in HCAR1KO, as this would implicate a more specific mechanisms as to how HCAR1 regulates these processes.

The authors discuss lactate as a key metabolic signal in the mechanism of HI injury, and in their discussion and graphical illustration, propose that endogenous lactate is indeed increased. Demonstrating lactate increase in their model would provide more support for this hypothesis (if this is possible). Also, Figure 2o-r data are from the intermediate zone only; when SVZ is analyzed (Supp Figure 1) there is a reduction of the neural progenitors in the HCAR1 KO, but not a reduction in the proliferative cells. I would expect the proliferation from the subventricular zone to be reduced in HCAR1 KO mice, and this could be discussed further in the manuscript. Also- are the data comparing contralateral proliferation and differentiation in WT and KO mice statistically compared, to ensure there is no baseline change to proliferation and neurogenesis on that side of the brain? Additionally, this study is very descriptive as it stands now. A rescue experiment in the HI-induced HCAR1 KO, potentially with one of the differentially expressed genes discovered in their RNA seq data set, would make a much stronger mechanistic case for lactate-HCAR1 dependent expression of genes directly involved in proliferative or differentiation processes. I think this type of rescue experiment is necessary to move this paper toward publication. Finally, There is also an opportunity to look at hippocampal DG proliferation and differentiation here, and if it is impaired in HCAR1 mice, there can be again an attempt for rescue. Justification for this experiment is further supported by the data in Figure 4 and Supp Figure 6 showing that HCAR1 KO mice show a transcriptional response to HI in the hippocampus, but not in the subventricular zone.

*Reviewer #2 (Recommendations for the authors):*

Kennedy et al. investigated the functions of lactate receptor HCAR1 towards tissue repair and associated neurogenesis in a neonatal mouse model of cerebral hypoxia-ischemia (HI). The authors successfully showed that they can mimic the neonatal HI pathology in mouse. They clearly demonstrated that compared to controls HCAR1 knockout mouse brains fail to recover post-insult, even after days. Using RNA sequencing, the authors also identified differentially modulated molecular pathways, predominantly affecting cell cycle regulation and tissue repair, indicating a role of HCAR1 as a transcriptional regulator. Finally, differences in the repair mechanisms of neurogenic niches in hippocampal and rostral subventricular zones were also demonstrated.

Although this study has significant translational impact, following concerns in experimental design and data interpretation were observed:

1. Normal expression pattern of HCAR1 mRNA and protein and their lack in the mutant brain has not been shown. Without this evidence, comparison across regions of interest will be incomplete.

2. Since the cell density changes significantly across some groups, the remaining proportion calculations (Figure 2P-R) should be done as a fraction of total number of cells, rather than area. Otherwise, the very relevance of differential cell densities is lost.

3. Figure 2K-N and Supp Figure 1A-D are identical images; authors should remove one set or assemble all the data to Figure 2. Further, if both IZ and SVZ have proliferating cells at postnatal day (P)9, other markers specifically demarcating IZ and SVZ should be used to check any kind of zonal expansion/reduction.

4. In postnatal and adult neurogenesis, the cell cycle period is short ranging from 17-22hours. Especially in injury models, S-phase and in turn total cycle in pathological conditions like injury/stroke is further shortened to ~3hrs and 12hrs respectively (PMID: 29765834). So, the scientific basis of performing multi-day BrdU injections with 24h gap to determine proliferation rate seems not convincing. Unless the authors can provide evidence/reasoning that the used paradigm is essential, it would be critical to repeat all BrdU proliferation experiments with a shorter 1-2hr BrdU pulse, and then reinterpreted for proliferation and differentiation (cell cycle exit).

5. It is unclear what is meant by "DCX+ and Ki67+ or BrdU^+^ cells". Ki67 marks all cell cycle phases while BrdU is specific to S-phase. These cannot be counted and analyzed together as they mark different cell populations. Also, no image depicting Ki67-staining was provided in the paper.

6. Regular neurogenesis and astrogliogenesis in hippocampus continue perinatally in mouse. As the mouse experiments are done at P9, how do the authors distinguish the normal hippocampal processes from insult-driven processes? Can any co-labeling differentiate these two? Also, Figure 1A demonstrates that large part of ipsilateral midline takes the TTC stain, hence is viable. Can that be a reason for less effect of ischemia in the hippocampus of this model?

7. Neurospheres were generated from P3 whole forebrains of control and mutant mice and analyzed ~10 days in vitro, as per Methods. And, in vivo proliferation/differentiation assays were done in P9 mice post-HI injury. However, the trend of significantly different proliferation and differentiation rates between control and mutant seen in vitro is not reproduced in vivo. What can be the scientific explanation? How will that impact the overall data interpretation?

8. Emphasis should be given in the Abstract, Introduction and Discussion that this work is on postnatal neurogenesis to avoid confusion. Some statements in Discussion are only true for adult neurogenesis, and not for neonatal mouse brain as used in this study. Portions thus should be reviewed and modified accordingly.

9. For cell counting, co-labeling with nuclear markers is preferable to filament markers (b-tubulin, GFAP).

10. Increasing the size of Figure 2 J may help in clarity.

11. Embryonically, neural stem cells are also called primary progenitors or radial glial cells. Later in development (neonatal/adults), there are Type B progenitors/precursors (mostly dedifferentiated astrocytes), intermediate Type C cells and Type A stages for both neuron and glia. Which cell type did the authors in this study refer to by the term "neural stem-progenitor cells"?

12. Lettering of figures (capital letters) and figure legends do not match.

13. Sentences in many Figure legends are incomplete.

14. It may help to show representative images of sham-treated control and mutant brains for better comparison.

15. As this is an injury model, inspecting apoptosis rate in the proliferative zones will be important.

16. Supplementary Figures 4,5: If FDR{less than or equal to}0.05 is the cut-off, the reason of showing all the non-significant GSEA groups is unclear.

*Reviewer #3 (Recommendations for the authors):*

Kennedy et al. examine the role of the specific lactate receptor HCAR1 in the response to neonatal HI, both in the short- (24h) and long- (42 day) term. Use of KO helps to clarify the importance of this receptor in tissue recovery, and use of microglial and neuronal staining, as well as in vitro and in vivo analyses, approach this analysis with different methods which strengthen the study. In addition the use of RNAseq suggest differences in the transcriptome of a particular region of the brain (SVZ) theorized to be heavily involved in the response to HI.

The authors have an interesting hypothesis and make a case for conclusive findings, although I would not say that they succeed. They do support the findings that more work needs to be done to understand the role of this receptor, and that it may play a critical role in response to injury, but there needs to be a more comprehensive examination of cell-type specific effects (including astrocytes and oligos), in other regions of the brain, and also the downstream signaling pathways need to be further evaluated to get at the mechanism of repair. For example, qPCR and protein quantification of specific factors and pathways. This is a good first step, but more detailed analyses are necessary to confirm.

If true, this would be an interesting finding in understanding injury and repair in the developing brain.

I will comment based on line/section of the manuscript.

47: I think your use of "stem cell" here and later in the manuscript is somewhat vague. You should probably stick to precursor or progenitor cell, or more clearly define the term and how you use it.

53: True that SVZ and SGZ are most well-described, but more recent evidence suggest role for local response to injury, also sub-pial and other regions.

82: Question – this model includes permanent occlusion of L CCA. What do you think is the role of reperfusion injury (as occurs in humans) on this particular receptor and response?

93: I'm not sure I understand the use of showing damage in individual sub-sections. You should consider including specific regions of the brain rather than a particular section, or remove altogether.

Section starting at 112: The use of in vitro neurospheres are interesting way to approach neurogenesis, where you show decreased proliferation at baseline. But the overall reduction in neurogenesis suggested by this manuscript don't conclusively show to me, if it is a decrease at baseline or a decrease in the response to injury, or both. Did you do OGD in vitro to see if there was also a decrease in the proportional response to injury?

131: How did you decide on BrdU on days 4-7? Why not start earlier? Also, a more thorough analysis of cell fate including cell types other than neurons and microglia would be useful, to see specific effects on immature/mature oligos and astrocytes after injury.

Section 179: Did you consider scRNAseq? I wonder if there are cell-type specific differences in transcription. Also, it is a fairly big leap to make any conclusions on the pathways based on bulk RNAseq alone. You need qPCR or protein quantification to draw more conclusions. Related, your wording in discussion needs to be softened in regards to mechanism without more conclusive evidence.

[Editors’ note: further revisions were suggested prior to acceptance, as described below.]

Thank you for resubmitting your work entitled "Lactate receptor HCAR1 regulates neurogenesis and microglia activation after neonatal hypoxia-ischemia" for further consideration by *eLife*. Your revised article has been evaluated by Marianne Bronner (Senior Editor) and a Reviewing Editor.

The manuscript has been improved but there are some remaining issues that need to be addressed, as outlined below:

- Please address each Reviewer's comments, as these are overall addressable issues. Most of these comments request clearer explanations in the manuscript text, or clarifications. Addressing these comments will greatly improve this manuscript, thank you.

*Reviewer #2 (Recommendations for the authors):*

This is a revised version of the manuscript that was previously submitted to *eLife*. Overall, compared to the original version, the authors largely addressed one major experimental suggestion and improved textual explanation of certain parts of the manuscript. However, majority of the remaining critical concerns remain inadequately addressed. There is also a mild unwanted tendency to fit the data interpretation as per some pre-fixed hypotheses/ideas. Hence some of the concerns that were pointed out in last review are reiterated here along with some new ones.

1. Figures 2,3,4, S1: BrdU marks cells undergoing DNA synthesis. As the authors mentioned, it is true that there are many BrdU protocols; however, each one has a logical basis for separate usage. The protocol that was used in this manuscript spans over 3-4 days, this will only provide information about the total number of cells that have undergone DNA synthesis during this period. This does NOT provide any information about the proliferation rate, unless other proliferation markers are also co-studied appropriately (like Hayashi et al. 2016, cited by the authors themselves). BrdU^+^ cells will exit cell-cycle within 3-12hrs (becoming non-proliferating); hence it is incorrect to state BrdU^+^ cells as "all proliferating cells" under the current experimental paradigm. This was already brought up in the last review, but was not satisfactorily addressed. Further, there is lack of clarity on the experimental motive – is this done to study proliferation rate or just to have a bigger BrdU^+^ cell pool?

2. Proper differentiation/cell exit assays that is required in order to comment on regulation of cell differentiation are lacking. Just calculating mature cells in a cohort (Figure 2I) is not a measure of differentiation. Additionally, GFAP is also a marker of early neural progenitors and glial precursors, not just a marker of mature astrocytes.

3. The authors proposed in the address to reviewers' comments that neurosphere assay may have several constraints that are not faced in vivo and hence they are giving different results. It is then unclear what biological relevance the authors want to show with the neurosphere assay data.

4. Cell density is a very important measurement, on which calculation of other cell proportions depend. Proportion of different cell types (Figure 2P-R) cannot be measured as absolute numbers when cell density is altered in a comparative group. This was also requested for reanalysis previously. Since the authors failed/were reluctant to perform a simple data reanalysis, here is a small illustrative example in the hope of getting the correct interpretation out:

Similar to the data scenario, suppose

WT brain: 6 total cells/unit area; 3 cell-type A/unit area

Mut brain: 6 total cells/unit area; 3 cell-type A/unit area

After HI,

WT(HI): become 10 total cells/unit area; 5 cell-type A/unit area

Mut(HI): remain at 6 total cells/unit area; 3 cell-type A/unit area

a. Analysis for cell-type A done as absolute number normalizing across area (like the manuscript): cell-type A in WT(HI) (5) is greater than that in all other groups (3).

b. Analysis for cell-type A done as proportion of total number of cells: All groups have same proportion of cell-type A (3/6=2 or 5/10=2).

The second type of analysis does not mask the effect of the data, as claimed by the authors; rather it is the only way to bring out the actual picture in view. This data analysis is required to be modified accordingly.

5. What was the landmark/marker used to precisely dissect striatal subventricular zone tissue for RNA sequencing? It is not clear why the RNA sequencing was done 3 days after HI while all the proliferation/neurogenesis experiments were performed 7days after HI despite hippocampus being damaged – any biological reason for this choice of ages?

6. There are multiple instances throughout the manuscript where data is interpreted without giving enough value to the related statistics, leading to misleading statements. Some examples are as follows, which need to be rectified:

Lines 122-123: The average size of HCAR1 KO spheres also tended to be smaller, although this was not statistically significant.

Lines 204-206: On the other hand, the contralateral hemisphere in HCAR1 KO mice was on average higher than in WT mice (albeit not statistically significant).

Lines 269-270: The average B1 levels were 78% lower in the HCAR1 KO compared with WT, but this result was not statistically significant (p=0.057)

*Reviewer #3 (Recommendations for the authors):*

This is a very interesting and important study testing the role of the lactate receptor HCAR1 in recovery after neonatal hypoxia-ischemia using a HCAR1 knockout. The authors have been thoughtful and thorough in their response to the previous reviews. They show significantly increased infarct size in the KO compared with wild-type animals, and then proceed to examine and demonstrate remarkable effects of the KO on processes likely to be involved in repair following injury: proliferation of neural progenitor cells, as well as astrocytes, oligodendrocytes, microglia; activation of microglia; and transcriptional activation in the subventricular zone. The authors are really to be commended for their wide-ranging analysis of the impact of HCAR1 KO. The study is largely descriptive, but this is not a liability as this is the foundation required for future studies. For example, given what they show, it would be of great interest to determine whether conditional knockout of HCAR1 in a particular cell type recapitulates the phenotype of the constitutive KO. One important issue is that the authors assume that deficits in repair are largely responsible for the differences in outcome between the WT and KO animals. However, in adult stroke, it is well-known that there is a penumbra of injured cells around the central infarcted area, and a great deal of work has been focused on the issue of saving the cells in the penumbra from being irreversibly committed to death. There is no mention of this issue in the manuscript, but it is an important problem, because it is conceivable that the effects of HCAR1 are entirely due to the protection of cells surviving the initial insult rather than the generation of new cells. At a minimum, the authors should address this issue in the discussion.

---

## [Author Response]

[Editors’ note: the authors resubmitted a revised version of the paper for consideration. What follows is the authors’ response to the first round of review.]

Reviewer #1 (Recommendations for the authors):This is a very interesting study implicating HCAR1, a G-protein coupled lactate receptor, in the pathophysiology of hypoxia-ischemia (HI)-induced impairments in brain tissue regeneration, cellular proliferation, neurogenesis and microglial activation. The authors use a HCAR1 KO mouse and provide convincing in vitro and in vivo evidence that newly proliferating cells in the intermediate zone (adjacent to SVZ) ipsilateral to the infarct/HI injury, as well as ipsilateral neurogenesis, is impaired after HI when HCAR1 is not present. Furthermore, microglial proliferation as well as activation is also reduced in the setting of HCAR1 KO. And finally, the authors provide RNA-sequencing data from HI-injured WT and HCAR1 KO mice, demonstrating that the significant differential gene expression occurring after HI in WT mice is not seen in HCAR1 KO mice. Overall, this study would be of major interest to scientists studying mechanisms of HI injury and intervention, possibly informing design of molecular targets to preserve cellular proliferation and tissue regeneration after ischemic injury.Strengths:The data presented here are very convincing that HCAR1 is required for tissue regeneration in the chronic period after HI injury (figure1). Figure 2 further shows evidence from both neurospheres and immunostaining that at the level of the intermediate zone, cellular proliferation and differentiation are directly impacted in the absence of HCAR1. Contralateral uninfarcted hemisphere is used as an intra-individual control, and very clearly demonstrates impact on proliferation and neurogenesis is specific to the hemisphere ipsilateral to the damage. It is interesting to note that HCAR1 absence alone does not impact proliferation and differentiation. Figure 3 shows that in the peri-infarct area, microglial activation and proliferation is repressed in HCAR1 KO mice. Figure 4 is descriptive in that it shows cell cycle pathway and complement pathway downregulation in HCAR1 KO SVZ from infarcted side of brain.

We thank the Reviewer for acknowledging that the study will be of major interest to the field and for pointing out that the data are very convincing in showing an effect of HCAR1 on tissue regeneration and cell proliferation after HI injury.

Weaknesses:There is not significance found in the proliferative assessment of the SVZ, in that both WT and HCAR1 mice have reduced proliferation (did not reach significance, however, the figure Supp 1F suggests a strong trend). It is also interesting that there was a transcriptional response to HI in hippocampus in HCAR1 KO but this was not explored further. It would be a logical next step, and one that would enhance the paper, to disclose what the downstream genes discovered from the RNA-seq data set that are involved in neurogenesis and innate immunity that are not activated in HCAR1KO, as this would implicate a more specific mechanisms as to how HCAR1 regulates these processes.The authors discuss lactate as a key metabolic signal in the mechanism of HI injury, and in their discussion and graphical illustration, propose that endogenous lactate is indeed increased. Demonstrating lactate increase in their model would provide more support for this hypothesis (if this is possible).

Elevated lactate in the ipsilateral hemisphere after HI has been demonstrated previously in a similar mouse HI model (Mikrogeorgiou et al., Dev Neurosci 2020, https://doi.org/10.1159/000506982). We have now added this point to the beginning of the discussion (page 11).

Also, Figure 2o-r data are from the intermediate zone only; when SVZ is analyzed (Supp Figure 1) there is a reduction of the neural progenitors in the HCAR1 KO, but not a reduction in the proliferative cells. I would expect the proliferation from the subventricular zone to be reduced in HCAR1 KO mice, and this could be discussed further in the manuscript.

We agree that an effect of HCAR1 was expected to be seen in the more basal VZ/SVZ immunolabeling. In the revised manuscript, we have increased the number of animals that were included in the analysis (Supp Figure 1). However, this did not lead to a statistically significant effect. There remains some controversy and disagreement in naming the SVZ, especially in the developing brain where the VZ is in transition from embryonic to the adult state. We believe both areas represent part of the sub-ventricular niche and have therefore renamed the areas to VZ/SVZ and SVZ/IZ, as has been done by several other investigators. Furthermore, Sejersted et al. 2011 (https://doi.org/10.1073/pnas.1106880108) reported the same findings using the exact same HImodel. As requested, we have now discussed this in the manuscript (page 12).

Also- are the data comparing contralateral proliferation and differentiation in WT and KO mice statistically compared, to ensure there is no baseline change to proliferation and neurogenesis on that side of the brain?

Yes, contralateral sides in KO versus WT were compared (described under Materials and methods in the section “Statistical Analysis”). There were no statistically significant differences between the contralateral hemispheres.

Additionally, this study is very descriptive as it stands now. A rescue experiment in the HI-induced HCAR1 KO, potentially with one of the differentially expressed genes discovered in their RNA seq data set, would make a much stronger mechanistic case for lactate-HCAR1 dependent expression of genes directly involved in proliferative or differentiation processes. I think this type of rescue experiment is necessary to move this paper toward publication.

We think that this experiment would be unlikely to succeed, given the high number of differentially expressed genes: we detected 6440 differentially expressed genes in the subventricular zone, of which 3182 were down regulated. Only related to the cell cycle, we found 53 down regulated genes. Therefore, we believe that upregulation of a single gene within this system would be highly unlikely to have a measurable effect. *We have been in contact with the editor, who allowed us to resubmit without addressing this point.*

Finally, there is also an opportunity to look at hippocampal DG proliferation and differentiation here, and if it is impaired in HCAR1 mice, there can be again an attempt for rescue. Justification for this experiment is further supported by the data in Figure 4 and Supp Figure 6 showing that HCAR1 KO mice show a transcriptional response to HI in the hippocampus, but not in the subventricular zone.

We agree that this would be interesting. Unfortunately, the hippocampus was completely degenerated on day 7 after HI (the time of fixation for immunostaining) in nearly all the mice that had undergone HI. Therefore, it was not possible to obtain comparable immunostaining data from the hippocampus. We have now mentioned this observation in the Results section (page 6). The reason we could still obtain RNA sequencing data from the hippocampus, was that tissue for RNAseq was isolated on day 3 after HI. We chose this difference in timing between RNAseq and immunostaining to be able to detect pathways that had been activated before most of the cell proliferation had taken place.

Reviewer #2 (Recommendations for the authors):Kennedy et al. investigated the functions of lactate receptor HCAR1 towards tissue repair and associated neurogenesis in a neonatal mouse model of cerebral hypoxia-ischemia (HI). The authors successfully showed that they can mimic the neonatal HI pathology in mouse. They clearly demonstrated that compared to controls HCAR1 knockout mouse brains fail to recover post-insult, even after days. Using RNA sequencing, the authors also identified differentially modulated molecular pathways, predominantly affecting cell cycle regulation and tissue repair, indicating a role of HCAR1 as a transcriptional regulator. Finally, differences in the repair mechanisms of neurogenic niches in hippocampal and rostral subventricular zones were also demonstrated.Although this study has significant translational impact, following concerns in experimental design and data interpretation were observed:

We appreciate that the reviewer acknowledges the significant translational impact of the study.

1. Normal expression pattern of HCAR1 mRNA and protein and their lack in the mutant brain has not been shown. Without this evidence, comparison across regions of interest will be incomplete.

mRNA expression of HCAR1, and its lack in the mutant brain, has been previously confirmed by us (Supplementary Table 1 in Morland et al., Nat Comm 2017, https://doi.org/10.1038/ncomms15557). Moreover, HCAR1 RFP reporter mice have been used to show HCAR1 expression in the brain, including the lining of the subventricular zone (Hadzic et al., Int J Mol Sci 2020, https://doi.org/10.3390/IJMS21186457 ).

Although several antibodies to HCAR1 are available commercially, none have proven to be specific (de Castro Abrantes, J Neurosci 2019, https://doi.org/10.1523/JNEUROSCI.2092-18.2019). Therefore, a further confirmation of HCAR1 protein expression in the different brain areas by immunostaining or western blot is not feasible.

2. Since the cell density changes significantly across some groups, the remaining proportion calculations (Figure 2P-R) should be done as a fraction of total number of cells, rather than area. Otherwise, the very relevance of differential cell densities is lost.

We disagree with this. Since we see that the total cell density increases after HI in WT, but not in KO (Figure 2O), we believe that showing the different cell types as a proportion of total cells instead of area would mask part of the effect.

3. Figure 2K-N and Supp Figure 1A-D are identical images; authors should remove one set or assemble all the data to Figure 2. Further, if both IZ and SVZ have proliferating cells at postnatal day (P)9, other markers specifically demarcating IZ and SVZ should be used to check any kind of zonal expansion/reduction.

We have removed the images in Supplementary Figure 1A-D, as requested.

There are, as far as we know, no clear consensus or markers to clearly define the layers of the SVZ and IZ. We agree that a more comprehensive analysis of the different cell populations and their density and possible zonal expansion would be interesting, but is in our opinion beyond the scope of this paper. We chose to count type A Neuroblasts as they are determined to a neuronal fate and thus tell us whether there is an effect on neurogenesis from HCAR1 before and after HI.

4. In postnatal and adult neurogenesis, the cell cycle period is short ranging from 17-22hours. Especially in injury models, S-phase and in turn total cycle in pathological conditions like injury/stroke is further shortened to ~3hrs and 12hrs respectively (PMID: 29765834). So, the scientific basis of performing multi-day BrdU injections with 24h gap to determine proliferation rate seems not convincing. Unless the authors can provide evidence/reasoning that the used paradigm is essential, it would be critical to repeat all BrdU proliferation experiments with a shorter 1-2hr BrdU pulse, and then reinterpreted for proliferation and differentiation (cell cycle exit).

Several different protocols exist for BrdU analyses. We are unsure of the exact experimental setup suggested by the reviewer and the reasoning for it.

1) If the reviewer here means that we should have performed a single 1-2hr BrdU pulse, we think this would have yielded a lower total density of countable proliferated cells compared with our setup with BrdU injections over several days. Therefore, our paradigm increases the strength of our BrdU analysis due to of the higher number of total counted cells.

2) If the reviewer means that we should have performed injections with 1-2hr intervals over several days, we agree that this would have increased our total number of quantified cells. Thus, this might have led us to detect even larger differences between KO and WT. So it is possible that our setup leads to some underestimation of the differences between KO and WT. However, such a setup would have put much more strain on the animals and could have led to unwanted stress responses.

Thus, we do not see any reason that our setup would lead to erroneous results. In support of our experimental setup, such (less frequent) multi day BrdU injections have been commonly used for similar purposes (Plane et al., Neurobiol Dis 2004, https://doi.org/10.1016/j.nbd.2004.04.003; Hayashi et al., Brain Res 2005 https://doi.org/10.1016/j.brainres.2004.12.048; Palibrk et al., Cell Death Dis 2016, https://doi.org/10.1038/cddis.2016.223).

5. It is unclear what is meant by “DCX+ and Ki67+ or BrdU^+^ cells”. Ki67 marks all cell cycle phases while BrdU is specific to S-phase. These cannot be counted and analyzed together as they mark different cell populations. Also, no image depicting Ki67-staining was provided in the paper.

We agree. The analysis was done in the previous manuscript due to shortage of BrdU treated samples, and the two groups were pooled together to reflect all proliferation both asymmetrically and symmetrical of the neuroblasts. We have now increased sample size, excluded the Ki67 samples and show only BrdU in this figure.

6. Regular neurogenesis and astrogliogenesis in hippocampus continue perinatally in mouse. As the mouse experiments are done at P9, how do the authors distinguish the normal hippocampal processes from insult-driven processes? Can any co-labeling differentiate these two?

We understand this question to be about the subventricular niche. The “basal” rate of peri/postnatal neurogenesis is determined by comparing the contralateral (undamaged) side between the genotypes. There were no statistically significant differences.

Also, Figure 1A demonstrates that large part of ipsilateral midline takes the TTC stain, hence is viable. Can that be a reason for less effect of ischemia in the hippocampus of this model?

Although the midline is viable, the hippocampus appeared severely injured at later stages and was often completely lost in our section on day 7 after HI (not shown, but now included in main text, page 6). Therefore, we do not believe that the hippocampus was less affected by ischemia compared with the SVZ. This is supported by the RNA sequencing data (from day 3 after HI), which shows a strong transcriptional response to HI in the hippocampus (see PCA plot in Suppl. Figure X and number of DEGs in Supplementary Table 1). Rather, the transcriptional response to HI in the hippocampus is similar between WT and HCAR1 KO mice, whereas in the SVZ there is a reduced response to HI in HCAR1 KO mice. Thus, the response to HI appears to be HCAR1 dependent in SVZ, but not in the hippocampus. These findings are supported by a recent publication showing an effect of HCAR1 on neurogenesis in the subventricular zone and not in the hippocampal subgranular zone (Lambertus et al. 2020, Acta Physiologica. https://doi.org/10.1111/apha.13587).

7. Neurospheres were generated from P3 whole forebrains of control and mutant mice and analyzed ~10 days in vitro, as per Methods. And, in vivo proliferation/differentiation assays were done in P9 mice post-HI injury. However, the trend of significantly different proliferation and differentiation rates between control and mutant seen in vitro is not reproduced in vivo. What can be the scientific explanation? How will that impact the overall data interpretation?

Neurospheres are isolated clusters of neural stem cells that lack the other components of the brain (including a vascular system). One reason for the different results between neurospheres and in vivo (contralateral hemispheres) could be compensatory mechanisms in vivo that are not present in vitro. Also, the process of creating the neurosphere assay from dissection, cell dissociation and culturing might cause a stress response in the cells that does not occur in the contralateral hemispheres in vivo. This has now been discussed in more detail in the manuscript (page 13).

8. Emphasis should be given in the Abstract, Introduction and Discussion that this work is on postnatal neurogenesis to avoid confusion. Some statements in Discussion are only true for adult neurogenesis, and not for neonatal mouse brain as used in this study. Portions thus should be reviewed and modified accordingly.

We agree. Changes have been made in the manuscript to clarify.

9. For cell counting, co-labeling with nuclear markers is preferable to filament markers (b-tubulin, GFAP).

We agree. However, we have chosen our antibodies based on our experience and their reliability. Except for neuroblasts, which often lie in clusters, we have used automatic counting using advanced image segmentation with WEKA-trainable segmentation tool and automatic thresholding algorithms combined with our own scripts, which exclude small objects/not whole cells and which are able to do overlap-analysis with DAPI and BrdU. Thus, our image analysis tools gives us the same possibility to count cells as one would with a nuclear marker. (If needed examples of the steps in image analysis can be provided).

10. Increasing the size of Figure 2 J may help in clarity.

This has been fixed.

11. Embryonically, neural stem cells are also called primary progenitors or radial glial cells. Later in development (neonatal/adults), there are Type B progenitors/precursors (mostly dedifferentiated astrocytes), intermediate Type C cells and Type A stages for both neuron and glia. Which cell type did the authors in this study refer to by the term “neural stem-progenitor cells”?

The term was written to include the in vivo and in vitro data we present. Our in vivo data present SVZ derived type A Neuroblasts (progenitors). Before differentiation, neurospheres consist mainly of neural stem and progenitor cells (NSPCs). We have now explained this point in a separate section under Materials and methods (page 20). For simplicity in the main text, we have termed the cells “progenitor cells” (as requested by Reviewer 3).

12. Lettering of figures (capital letters) and figure legends do not match.13. Sentences in many Figure legends are incomplete.

Fixed.

14. It may help to show representative images of sham-treated control and mutant brains for better comparison.

As the contralateral hemisphere of our hypoxic-ischemia treated animals has been presented and validated as a control previously (Sejerstad et al., PNAS 2011, https://doi.org/10.1073/pnas.1106880108), we have not performed sham treatment for our injury analyses (Figure 1) or immunohistochemistry experiments (Figure 2,3,4).

15. As this is an injury model, inspecting apoptosis rate in the proliferative zones will be important.

Our data suggest that the major difference between the genotypes lies in the regeneration and innate immune response rather than in initial cell death and apoptosis. This is supported by the measurements of acute injury (Figure 1A) as well as the RNA sequencing from SVZ three days post injury, in which apoptosis was not among the altered pathways (Figure 5B). Although it could still be interesting to inspect apoptosis rates directly, we feel that this is not necessary for the conclusions in this manuscript.

16. Supplementary Figures 4,5: If FDR{less than or equal to}0.05 is the cut-off, the reason of showing all the non-significant GSEA groups is unclear.

Although the data are not statistically significant with our cut-off, they show tendencies that could have biological significance. We have therefore chosen to keep them in the supplement.

Reviewer #3 (Recommendations for the authors):Kennedy et al. examine the role of the specific lactate receptor HCAR1 in the response to neonatal HI, both in the short- (24h) and long- (42 day) term. Use of KO helps to clarify the importance of this receptor in tissue recovery, and use of microglial and neuronal staining, as well as in vitro and in vivo analyses, approach this analysis with different methods which strengthen the study. In addition the use of RNAseq suggest differences in the transcriptome of a particular region of the brain (SVZ) theorized to be heavily involved in the response to HI.The authors have an interesting hypothesis and make a case for conclusive findings, although I would not say that they succeed. They do support the findings that more work needs to be done to understand the role of this receptor, and that it may play a critical role in response to injury, but there needs to be a more comprehensive examination of cell-type specific effects (including astrocytes and oligos), in other regions of the brain, and also the downstream signaling pathways need to be further evaluated to get at the mechanism of repair. For example, qPCR and protein quantification of specific factors and pathways. This is a good first step, but more detailed analyses are necessary to confirm.If true, this would be an interesting finding in understanding injury and repair in the developing brain.I will comment based on line/section of the manuscript.47: I think your use of "stem cell" here and later in the manuscript is somewhat vague. You should probably stick to precursor or progenitor cell, or more clearly define the term and how you use it.

Agreed. We have now changed the text to say “progenitor cell” throughout the manuscript. In addition, we have added a small section under Materials and methods (page 20) where we specify the types of progenitors in vivo and in vitro.

53: True that SVZ and SGZ are most well-described, but more recent evidence suggest role for local response to injury, also sub-pial and other regions.

We have now changed the wording in the introduction as to not exclude other neurogenic areas and we have mentioned these in the discussion (page 12).

82: Question – this model includes permanent occlusion of L CCA. What do you think is the role of reperfusion injury (as occurs in humans) on this particular receptor and response?

Recent data from adult rats suggest that HCAR1 is not involved in the effects of lactate on reperfusion injury (Buscemi et al., Front Physiol 2021, https://doi.org/10.3389/fphys.2021.689239), but this has not been tested in neonatal HI as far as we know. We have now discussed this topic in the manuscript (page 12).

93: I'm not sure I understand the use of showing damage in individual sub-sections. You should consider including specific regions of the brain rather than a particular section, or remove altogether.

We agree that the sub-sections figure is not so informative. Unfortunately the large damage on the ipsilateral side makes it near-impossible to measure specific regions on this side of the brain. Some regions, especially the hippocampus is completely missing 42 days after HI in many sections. Therefore, we have chosen to remove the measurements of sub-sections from Figure 1. We mention the loss of hippocampus in the text, but have not quantified this, for the reason explained here.

Section starting at 112: The use of in vitro neurospheres are interesting way to approach neurogenesis, where you show decreased proliferation at baseline. But the overall reduction in neurogenesis suggested by this manuscript don't conclusively show to me, if it is a decrease at baseline or a decrease in the response to injury, or both. Did you do OGD in vitro to see if there was also a decrease in the proportional response to injury?

We have not done OGD in vitro. The discrepancy between the neurosphere data and the in vivo control data (contralateral hemispheres) could be due to compensatory mechanisms in vivo or possibly (unintended) stress to the cells caused by the neurosphere protocol. We have now discussed this in the manuscript (page 13) and in the response to reviewer #2 above.

131: How did you decide on BrdU on days 4-7? Why not start earlier? Also, a more thorough analysis of cell fate including cell types other than neurons and microglia would be useful, to see specific effects on immature/mature oligos and astrocytes after injury.

Various protocols exist for BrdU injections after ischemic injury, with the starting point mostly varying between one and five days after the induced injury. One argument for starting around day 35 post injury is to avoid BrdU incorporation into cells that are taking up BrdU due to DNA repair (rather than proliferation, Ong et al., Pediatr Res 2005, https://doi.org/10.1203/01.PDR.0000179381.86809.02; Cooper-Kuhn and Kuhn, Dev Brain Res 2002, https://doi.org/10.1016/S0165- 3806(01)00243-7), which presumably occurs more frequently in the first days after the injury. Nevertheless, we agree that starting earlier with injections might have shown larger differences between WT and HCAR1 KO mice since more cells would have been included in the analyses. Thus, our data may underestimate the effect of HCAR1 on ischemia induced proliferation.

We thank the reviewer for suggesting to include oligodendrocytes and astrocytes in the analyses. We have now performed these experiments (Figure 4). We find a significant increase in the proliferation of astrocytes and oligodendrocytes in WT, but not in HCAR1 KO mice. We believe these data further strengthen the current study by underlining the role of HCAR1 in promoting overall cell proliferation after HI.

Section 179: Did you consider scRNAseq? I wonder if there are cell-type specific differences in transcription. Also, it is a fairly big leap to make any conclusions on the pathways based on bulk RNAseq alone. You need qPCR or protein quantification to draw more conclusions. Related, your wording in discussion needs to be softened in regards to mechanism without more conclusive evidence.

Although single cell RNA sequencing would be very interesting to investigate the cell-specific effects of HCAR1, this is beyond the scope of the current study. Given that we have now analysed proliferation of different glial cells by immunolabelling (previous point), we feel that we have partly addressed the cell-specific effects of HCAR1. We have been in contact with the editor, who allowed us to resubmit without performing scRNAseq.

As suggested, we have now confirmed the changes in some cell cycle genes on protein level, by western blotting (Figure 5F-H). In addition, we have softened the claims in the discussion by mentioning the need for further confirmation on the protein level (bottom of page 13).

[Editors’ note: what follows is the authors’ response to the second round of review.]

The manuscript has been improved but there are some remaining issues that need to be addressed, as outlined below:Reviewer #2 (Recommendations for the authors):This is a revised version of the manuscript that was previously submitted to eLife. Overall, compared to the original version, the authors largely addressed one major experimental suggestion and improved textual explanation of certain parts of the manuscript. However, majority of the remaining critical concerns remain inadequately addressed. There is also a mild unwanted tendency to fit the data interpretation as per some pre-fixed hypotheses/ideas. Hence some of the concerns that were pointed out in last review are reiterated here along with some new ones.1. Figures 2,3,4, S1: BrdU marks cells undergoing DNA synthesis. As the authors mentioned, it is true that there are many BrdU protocols; however, each one has a logical basis for separate usage. The protocol that was used in this manuscript spans over 3-4 days, this will only provide information about the total number of cells that have undergone DNA synthesis during this period. This does NOT provide any information about the proliferation rate, unless other proliferation markers are also co-studied appropriately (like Hayashi et al. 2016, cited by the authors themselves). BrdU^+^ cells will exit cell-cycle within 3-12hrs (becoming non-proliferating); hence it is incorrect to state BrdU^+^ cells as "all proliferating cells" under the current experimental paradigm. This was already brought up in the last review, but was not satisfactorily addressed. Further, there is lack of clarity on the experimental motive – is this done to study proliferation rate or just to have a bigger BrdU^+^ cell pool?

Our motive to use the current BrdU protocol was to compare density of proliferated cells after HI between WT and HCAR1 KO mice. If we understand the reviewer correctly, the issue is not with our BrdU protocol, but with our wording, where we inaccurately had used the terms “proliferation rate” and “all proliferating cells” at two places in the manuscript. We have now replaced “Reduced proliferation rate” with “less proliferation” (line 123). “All proliferating cells” is replaced with “proliferated cells” (Figure legend for Figure 2).

2. Proper differentiation/cell exit assays that is required in order to comment on regulation of cell differentiation are lacking. Just calculating mature cells in a cohort (Figure 2I) is not a measure of differentiation. Additionally, GFAP is also a marker of early neural progenitors and glial precursors, not just a marker of mature astrocytes.

We have softened our wording throughout the manuscript such that we no longer claim an effect of HCAR1 on differentiation (e.g. line 131). We have also changed the wording to say GFAP+ cells rather than astrocytes and have mentioned the fact that GFAP also labels early neural progenitors (Figure legend for Figure 2 and line 127)

3. The authors proposed in the address to reviewers' comments that neurosphere assay may have several constraints that are not faced in vivo and hence they are giving different results. It is then unclear what biological relevance the authors want to show with the neurosphere assay data.

The neurospheres offer a simplified and isolated in vitro system where proliferation, self-renewal and differentiation can be tested in a controlled environment. This is now mentioned in the manuscript (line 119).

4. Cell density is a very important measurement, on which calculation of other cell proportions depend. Proportion of different cell types (Figure 2P-R) cannot be measured as absolute numbers when cell density is altered in a comparative group. This was also requested for reanalysis previously. Since the authors failed/were reluctant to perform a simple data reanalysis, here is a small illustrative example in the hope of getting the correct interpretation out:Similar to the data scenario, supposeWT brain: 6 total cells/unit area; 3 cell-type A/unit areaMut brain: 6 total cells/unit area; 3 cell-type A/unit areaAfter HI,WT(HI): become 10 total cells/unit area; 5 cell-type A/unit areaMut(HI): remain at 6 total cells/unit area; 3 cell-type A/unit areaa. Analysis for cell-type A done as absolute number normalizing across area (like the manuscript): cell-type A in WT(HI) (5) is greater than that in all other groups (3).b. Analysis for cell-type A done as proportion of total number of cells: All groups have same proportion of cell-type A (3/6=2 or 5/10=2).The second type of analysis does not mask the effect of the data, as claimed by the authors; rather it is the only way to bring out the actual picture in view. This data analysis is required to be modified accordingly.

We have now performed the requested analyses (Figure 2P-R, Figure 3F-G and Figure 4E-Q, and results are described in the main text).

The results are mostly the same as previously, except for in oligodendrocytes, where there is no longer a significant increase in Olig2 cells after HI and no difference between WT and HCAR1 KO. Thus, the main conclusion of the study remains the same.

However, we do not agree with the reviewer that this is the only way to bring out the actual picture. Cells/DAPI (as requested by the reviewer) will say how the labelled cell type (e.g. Olig2 cells) changed in comparison with all cells (DAPI), whereas cells/unit area (our “old” analyses) will say how this cell type changed overall (independently of all the other cells). Thus, for Olig2 cells, we see that these cells increased in overall density (cells/unit area) after HI, but since all cells (i.e. DAPI+ cells) increased equally much, olig2/DAPI is not increased. We still think that the overall density of a cell type physiologically relevant. We have therefore chosen to include our old analysis as supplement to Figures2-4 and have included a discussion of the differences between the two analyses (paragraph starting on line 359 of the discussion). Of note, our “old” analysis, now in the supplement, is commonly used to study neurogenesis (see for instance Hoshi et al., Science Advances 2021 https://doi.org/10.1126/sciadv.abj8080 or Lambertus et al., Acta Physiologica 2021 https://doi.org/10.1111/apha.13587).

5. What was the landmark/marker used to precisely dissect striatal subventricular zone tissue for RNA sequencing? It is not clear why the RNA sequencing was done 3 days after HI while all the proliferation/neurogenesis experiments were performed 7days after HI despite hippocampus being damaged – any biological reason for this choice of ages?

The Corpus callosum and the shape of the lateral ventricles were used as landmarks for SVZ dissection. A detailed description of the SVZ dissection has now been included under “RNA sequencing and analysis” in the Methods section.

We chose to perform immunostaining 7 days after HI because we wanted to detect proliferating cells between days 4 and 7 after HI by BrdU injections. The rationale for this timing was partly to avoid BrdU incorporation into cells that are taking up BrdU due to DNA repair (rather than proliferation, Ong et al., 2005 DOI: 10.1203/01.PDR.0000179381.86809.02; Cooper-Kuhn and Kuhn 2002 https://doi.org/10.1016/S0165-3806(01)00243-7), which presumably occurs more frequently in the first days after the injury. Moreover, cell proliferation after HI is high 3-7 days after HI (Plane et al., Neurobiol of Disease, 2004 https://doi.org/10.1016/j.nbd.2004.04.003; Velthoven et al., Brain, Behavior, and Immunity 2010 https://doi.org/10.1016/j.bbi.2009.10.017). Thus, since BrdU injections started already on Day 4 after HI, the immunostaining would be able to capture the cell proliferations occurring already at that stage.

The rationale for performing RNAseq at day 3, was to try and capture the cellular pathways that were initiated at the start of the brain tissue regeneration process, in the sub-acute phase of HI.

6. There are multiple instances throughout the manuscript where data is interpreted without giving enough value to the related statistics, leading to misleading statements. Some examples are as follows, which need to be rectified:Lines 122-123: The average size of HCAR1 KO spheres also tended to be smaller, although this was not statistically significant.Lines 204-206: On the other hand, the contralateral hemisphere in HCAR1 KO mice was on average higher than in WT mice (albeit not statistically significant).Lines 269-270: The average B1 levels were 78% lower in the HCAR1 KO compared with WT, but this result was not statistically significant (p=0.057)

The above mentioned statements have been removed.

Reviewer #3 (Recommendations for the authors):This is a very interesting and important study testing the role of the lactate receptor HCAR1 in recovery after neonatal hypoxia-ischemia using a HCAR1 knockout. The authors have been thoughtful and thorough in their response to the previous reviews. They show significantly increased infarct size in the KO compared with wild-type animals, and then proceed to examine and demonstrate remarkable effects of the KO on processes likely to be involved in repair following injury: proliferation of neural progenitor cells, as well as astrocytes, oligodendrocytes, microglia; activation of microglia; and transcriptional activation in the subventricular zone. The authors are really to be commended for their wide-ranging analysis of the impact of HCAR1 KO. The study is largely descriptive, but this is not a liability as this is the foundation required for future studies. For example, given what they show, it would be of great interest to determine whether conditional knockout of HCAR1 in a particular cell type recapitulates the phenotype of the constitutive KO. One important issue is that the authors assume that deficits in repair are largely responsible for the differences in outcome between the WT and KO animals. However, in adult stroke, it is well-known that there is a penumbra of injured cells around the central infarcted area, and a great deal of work has been focused on the issue of saving the cells in the penumbra from being irreversibly committed to death. There is no mention of this issue in the manuscript, but it is an important problem, because it is conceivable that the effects of HCAR1 are entirely due to the protection of cells surviving the initial insult rather than the generation of new cells. At a minimum, the authors should address this issue in the discussion.

We thank the reviewer for the positive comments on our manuscript and for good suggestions for improvements. We have now included a paragraph about apoptosis in the discussion (starting at line 397).